# Colocalization Analysis of Peripheral Myelin Protein-22 and Lamin-B1 in the Schwann Cell Nuclei of Wt and TrJ Mice

**DOI:** 10.3390/biom12030456

**Published:** 2022-03-16

**Authors:** María Vittoria Di Tomaso, Lucía Vázquez Alberdi, Daniela Olsson, Saira Cancela, Anabel Fernández, Juan Carlos Rosillo, Ana Laura Reyes Ábalos, Magdalena Álvarez Zabaleta, Miguel Calero, Alejandra Kun

**Affiliations:** 1Departamento de Genética, Instituto de Investigaciones Biológicas Clemente Estable, Montevideo 11600, Uruguay; olsson.daniela@gmail.com (D.O.); saicancela@gmail.com (S.C.); areyes@fcien.edu.uy (A.L.R.Á.); magda28963@gmail.com (M.Á.Z.); 2Laboratorio de Biología Celular del Sistema Nervioso Periférico, Departamento de Proteínas y Ácidos Nucleicos, Instituto de Investigaciones Biológicas Clemente Estable, Montevideo 11600, Uruguay; lvazquez@iibce.edu.uy; 3Laboratorio de Neurobiología Comparada, Departamento de Neurociencias Integrativas, Instituto de Investigaciones Biológicas Clemente Estable, Montevideo 11600, Uruguay; afernandez@iibce.edu.uy (A.F.); jrosillo@iibce.edu.uy (J.C.R.); 4Laboratorio de Neurociencias, Facultad de Ciencias, Universidad de la República, Montevideo 11400, Uruguay; 5Departamento de Histología y Embriología, Facultad de Medicina, Universidad de la República, Montevideo 11800, Uruguay; 6Unidad de Microscopía Electrónica de Barrido, Universidad de la República, Montevideo 11400, Uruguay; 7Instituto de Salud Carlos III, Centro de Investigación Biomédica en Red de Enfermedades Neurodegenerativas (CIBERNED), Unidad de Encefalopatías Espongiformes (UFIEC), 28029 Madrid, Spain; mcalero@isciii.es; 8Queen Sofia Foundation Alzheimer Center, CIEN Foundation, 28031 Madrid, Spain; 9Sección Bioquímica, Facultad de Ciencias, Universidad de la República, Montevideo 11400, Uruguay

**Keywords:** PMP22, Lamin B1, Trembler-J, Schwann cells nuclei, colocalization-analysis

## Abstract

Myelination of the peripheral nervous system requires Schwann cells (SC) differentiation into the myelinating phenotype. The peripheral myelin protein-22 (PMP22) is an integral membrane glycoprotein, expressed in SC. It was initially described as a growth arrest-specific (*gas3*) gene product, up-regulated by serum starvation. PMP22 mutations were pathognomonic for human hereditary peripheral neuropathies, including the Charcot-Marie-Tooth disease (CMT). Trembler-J (TrJ) is a heterozygous mouse model carrying the same *pmp22* point mutation as a CMT1E variant. Mutations in lamina genes have been related to a type of peripheral (CMT2B1) or central (autosomal dominant leukodystrophy) neuropathy. We explore the presence of PMP22 and Lamin B1 in Wt and TrJ SC nuclei of sciatic nerves and the colocalization of PMP22 concerning the silent heterochromatin (HC: DAPI-dark counterstaining), the transcriptionally active euchromatin (EC), and the nuclear lamina (H3K4m3 and Lamin B1 immunostaining, respectively). The results revealed that the number of TrJ SC nuclei in sciatic nerves was greater, and the SC volumes were smaller than those of Wt. The myelin protein PMP22 and Lamin B1 were detected in Wt and TrJ SC nuclei and predominantly in peripheral nuclear regions. The level of PMP22 was higher, and those of Lamin B1 lower in TrJ than in Wt mice. The level of PMP22 was higher, and those of Lamin B1 lower in TrJ than in Wt mice. PMP22 colocalized more with Lamin B1 and with the transcriptionally competent EC, than the silent HC with differences between Wt and TrJ genotypes. The results are discussed regarding the probable nuclear role of PMP22 and the relationship with TrJ neuropathy.

## 1. Introduction

The 22 kDa PMP22 is a Schwann cell (SC) highly expressed protein, being one of the main components of the peripheral nervous system (PNS) compact myelin [1,2,3,4].

The PMP22 gene was detected primarily in mouse NIH 3T3 fibroblasts as a growth arrest-specific (*gas3*) gene up-regulated by serum starvation [5]. A CD25 [6] and a SR13 [7] cDNA encoding transcripts were isolated from rat sciatic nerves and both revealed down-regulation after sciatic nerve injury [6,7]. The CD25 nucleotide sequence was identical to the *gas3*, and the transcript predicted a transmembrane protein [6]. The SR13 encoded a 22-kDa myelin glycoprotein localized in the myelin sheath of the sciatic nerves and expressed exclusively in the PNS [1] with homology to PAS-II, a protein isolated from bovine peripheral myelin [7,8]. Hence, based on the sequence identity among SR13, CD25 and PAS-II, the name PMP22 was proposed for these proteins [1].

It was demonstrated in cultured SC, that PMP22 is first synthesized as an 18 kDa precursor protein and then post-translationally modified by N-linked glycosylation in the endoplasmic reticulum and Golgi compartments [9]. PMP22 has a rapid turnover rate (half-life of 30–60 min), and only a small proportion of the newly synthesized glycoprotein reaches the plasma membrane [9,10]. The resulting 22 kDa PMP22 is a highly hydrophobic integral membrane protein of 160 residues, which detailed structure remains elusive. As of today, four putative transmembrane (TM) domains (TM1, TM2, TM3 and TM4) have been described [5,6,11].

The *pmp22*, located at the 17p11.2 human chromosome region, consists of six exons conserved in humans and rodents. In rats and mice, the first alternatively transcribed exon, originating two distinct transcripts (Exon-1A and Exon-1B) that differ only in their 5’ untranslated promoter (P1A and P1B) regions [12,13]. P1A showed homology with promoters of tissue-specific genes, and P1B with promoters of housekeeping genes [14]. Exon-1A (CD25) is expressed predominantly in the SC of the sciatic nerve [1,15], but Exon-1B (SR13) is also transcribed at the central nervous system (CNS) and non-neural tissues, at a relatively low level [14,15,16]. Three transcripts, regulated by alternatively used promoters, were described in humans. P1A showed homology to tissue-specific gene promoters, while PIB and P1C showed homology to housekeeping gene promoters. All three transcripts are expressed in sciatic nerves, showing more widespread biological variability between tissues [17].

*Pmp22* mutations were pathognomonic for the main peripheral hereditary neuropathies Charcot-Marie-Tooth disease (CMT). The analysis of the pathological mechanisms of the peripheral neuropathies contributed to associating PMP22 function with the structure, sheath formation, maintenance, and stability of the myelin, and with the reduction of internode capacitance to facilitate saltatory nerve conduction [18,19].

Charcot-Marie-Tooth disease represents the most common human neuromuscular disorder comprising several genetically and clinically heterogeneous hereditary neuropathies. They are classically classified as: primarily demyelinating or demyelinating type neuropathy (CMT1) and primarily axonal degeneration or axonal type neuropathy (CMT2), also divided into different sub-types [20,21,22].

The most frequent myelinopathy, CMT1A sub-type, has been genetically linked with the duplication of the human chromosome 17p11.2-p12 region, containing the *pmp22*. The mutation originates a *pmp22* trisomy which associates with an over-expression of PMP22 [23,24,25]. On the contrary, the HNPP (Hereditary Neuropathy with liability to Pressure Palsies) is associated with a deletion in the 17p11.2 region, generating a *pmp22* monosomy, which results in an under-expression of PMP22 [26,27]. Furthermore, PMP22 point mutations originate the clinical CMT1E phenotype, a subtype of CMT1 [19,28,29].

The Trembler (Tr) and Trembler-J (TrJ) are mice models carrying different point mutations in the *pmp22* (G150D or L16P, respectively) that display pathological phenotypes resembling those of CMT1E patients [13,28,29]. In TrJ mice, the substitution L16P affects the first transmembrane domain (TM1) of PMP22, interfering with myelination [14,30,31,32,33,34,35,36,37]. Viable individuals are those carrying the mutation in heterozygosis (TrJ/+), meanwhile, while homozygous TrJ/TrJ genotype leads to severe neuropathy with mice death at 17–18 days of age [33,37]. The TrJ phenotype has been related to the failure of PMP22 to fold and transport from the endoplasmic reticulum to the plasma membrane, forming heterodimers with wild-type PMP22, in cytoplasmic aggresomes [38,39].

Peripheral myelination requires SC differentiation into the myelinating phenotype [40,41,42]. In addition to its myelinating function, it was proposed that PMP22 could also be involved in cell cycle control, maintaining the G0 state of the differentiated SC [5,43].

Certain CMT diseases are associated with mutations in other proteins related to nuclear structure/function. Such is the case of CMT2-B1, a laminopathy caused by a point mutation in the Lamin A (LMNA) gene. LMNA encodes the Lamin A/C component of the nuclear lamina, a protein network composed of type V intermediate filament proteins, underlying the inner nuclear envelope in metazoan cells [44,45]. Also, the Adult-Onset-Autosomal Dominant Leukodystrophy (ADLD), a central demyelination neuropathy, is produced by a heterozygous tandem genomic duplication of the Lamin B1 gene (*LMNB1*), codifying for another component of the nuclear lamina. The over-expression of Lamin B1 is associated with a symmetric demyelination of the brain and spinal cord [46,47].

In the current approach, we quantified the expressions of PMP22 and Lamin B1 in the SC nuclei of the heterozygous mouse model CMT1E (TrJ) and its counterpart (Wt). We examined the colocalization of PMP22 with the Lamin B1 and SC nuclear subcompartments as a first approach to explore its putative function in SC nuclei. We observed and reported for the first time that the expression level of Lamin B1 was lower in SC nuclei of TrJ than in Wt, colocalizing with PMP22 in both genotypes. Furthermore, a higher expression of PMP22 was observed in the nuclei of TrJ SC compared to those of Wt. Although in both genotypes, PMP22 was colocalized more with transcriptionally active euchromatin than with silent heterochromatin, the level of colocalization was lower in TrJ than in Wt. The general results suggest an alteration in the eventual nuclear function of the mutated PMP22.

## 2. Materials and Methods

### 2.1. Experimental Animals

The analysis was performed in 15 TrJ male mice carrying the T1703C point mutation (L16P: leucine for proline substitution at position 16 of PMP22), acquired from the B6.D2-Pmp22<Tr-J>/J background (Jackson Laboratories, Bar Harbor, ME, USA) and in 15 wild Wt male mice. The procedures were approved by the institutional Ethics Committee (CEUA-IIBCE, MEC, Uruguay). All experiments were carried out in strict accordance with the relevant regulations and guidelines (Uruguayan Law number 18611).

Five-month-old mice were housed at the IIBCE animal facility in a controlled environment (12:12 light-dark cycles) and a temperature of 21 ± 3 °C with free access to food and water. Mice were numbered by the method of ear punching [48].

### 2.2. Mice Phenotyping

TrJ were phenotypically distinguished from Wt mice by suspending the mice by their tails; unlike Wt, TrJ mice cannot open their hind limbs (MSTT procedure) [49].

### 2.3. Whole Sciatic Nerve Fibers Preparations

Wt and TrJ mice were euthanized by cervical dislocation according to AVMA [50]. Immediately after, Wt and TrJ mice were placed in the ventral decubitus position and sprayed with 70% ethanol. The skin of each leg was removed, and the thigh muscles were separated. A closed scissor was introduced in the aponeurosis, parallel to the femur, and slowly opened to slit the aponeurosis at the knee and hip. The sciatic nerve visualized parallel to the bone was cut, first at the knee level and then at the entry of the hip. Once the sciatic nerve was obtained, it was fixed by immersion in 3% paraformaldehyde (PFA) in PHEM buffer (60 mM PIPES, 25 mM HEPES, 10 mM EGTA, 2 mM MgCl2; pH 7.2–7.4), and kept at 4 °C. The fixation process was extended up to 1 h, at 4 °C, with stirring in an orbital shaker. After that, the sciatic nerve fibers were washed in PHEM (30 min, every 5 min). The epineurium was removed, and nerve fibers were slightly “teased” under a stereoscopic loupe and subsequently immunostained [51].

### 2.4. Immunolabelling of H3K4m3, PMP22 and Lamin B1 in Schwann Cell Nuclei

An unmasking process was accomplished by immersion (10 s) of Wt and TrJ sciatic nerve fibers in a 70% formic acid solution (freshly prepared) and quickly washed, first in water, and then in PHEM buffer. The nerves were then permeabilized (30 min, 37 °C) in 0.1% Triton-X 100-PHEM solution and next, incubated simultaneously with the specific antibodies rabbit polyclonal or mouse monoclonal antibodies anti-PMP22 (Cat# ab61220, RRID: AB_ 944897, Abcam, Cambridge, UK, or Cat# sc65739 RRID: AB_2167002, Santa Cruz Biotechnology, Middlesex University, Wembley, UK, respectively), mouse monoclonal anti-H3K4m3 (Cat# ab12209, RRID: AB_442957, Abcam, Cambridge, UK), and Lamin B1(Cat# ab16048, RRID: AB_10107828, Abcam, Cambridge, UK) first at 37 °C, 1 h, and then at 4 °C, 24 h. The dilution for the antibodies was 1:100 in IB (0.1% bovine seroalbumin, 0.15 mM glycine in PHEM buffer, pH 7.4). 

Subsequently, nerve fibers were washed three times in IB, at room temperature, 5 min each, and then, nonspecific binding was blocked by incubating in BB (5% normal goat serum, 0.1% bovine serum albumin, 0.15 mM glycine in PHEM buffer, pH 7.4) for 30 min at 37 °C. Immediately, the nerve fibers were incubated 45 min in darkness, at 37 °C with the secondary antibodies: goat anti-rabbit IgG (H&L), Alexa Fluor 488 conjugated (Cat# ab150077, RRID: AB_2630356, Abcam, Cambridge, UK), goat anti-rabbit IgG (H&L)) Alexa Fluor 546 conjugated (Cat# A11035, RRID: AB_141370, Molecular Probes, Eugene, OR, USA) or goat anti-chicken IgY (H + L) Alexa Fluor 633 conjugated (Cat# A-21103, RRID: AB_2535756, Thermo Fisher Scientific, Waltham, MA, USA). 

The dilution for the anti-antibodies was 1:1000 in IB. After that, nerve fibers were washed two times in IB, 5 min each, at 37 °C, and three times in PHEM, 5 min each, at room temperature, always in darkness. Then, they were counterstained with DAPI (Cat# D1306, RRID: AB_2629482), Thermo Fisher Scientific, Waltham, MA, USA), dilution 1:1000 in IB, at 37 °C, for 45 min in darkness. All the procedures were performed under free-floating conditions and agitation. Finally, the sciatic nerve fibers were teased over coverslips and mounted in ProLong Gold antifade (P36930, Invitrogen, Eugene, OR, USA) and dried in darkness, at room temperature 24–48 h. Then, the preparation where stored at 4 °C, until confocal microscopy analysis.

The trimethylation of the fourth lysine residue of the histone H3 core (H3K4me3) represents an epigenetic mark of EC, the Lamin B1 is a component of the nuclear lamina, and the lamina 4,6-diamidino-2-phenylindole (DAPI) is a fluorescent dye that binds selectively to double-stranded DNA, preferentially at AT-reach regions.

### 2.5. Confocal Microscopy Procedure

Wt and TrJ sciatic nerve fibers preparations were visualized in parallel using a Zeiss LSM 800 confocal microscope with an air scan module. Confocal z-stack images were acquired using a Plan Apo N 60× oil NA 1.42 oil immersion lens and 2× magnifications. Additionally, low magnification fields were captured using a 40× Plan-Apochromatic lens. In all cases, images were acquired employing 1024 × 1024 pixel format (voxel size: ∆x/∆y/∆z = 0.379/0.379/1.00 µm)

The presence of autofluorescence or nonspecific reactivity was ruled out, taking as reference a non-immunostained sample or a sample incubated without specific primary antibodies, respectively. To adjust the ratio signal/background noise, the photomultiplier laser maximal levels of each channel were fixed, taking as reference a negative control prepared without specific antibodies. The maximal level was fixed until the appearance of a few brilliant non-specific signals. All sciatic nerve fiber images were obtained at the same initial settings to ensure comparable analysis conditions. The z-stack capture, with z-step distances of 0.21 microns, was done using the confocal microscopy software, fixing the bottom position, when the image of the object of interest appeared (first slice image) and the top position, when the image of the object of interest disappeared (last slice image).

### 2.6. Image Analysis

#### 2.6.1. Quantification of Sciatic Fiber and Schwann Cell Nuclei

On 40× images of TrJ and Wt sciatic nerves, the numbers of fibers, SC nuclei per fiber were estimated on DAPI channel of each z-stack image using the Fiji image processing free software. Data on the number of sciatic nerve fibers were obtained by analyzing images of TrJ fibers in focal planes covering areas equivalent to those analyzed in Wt. All images were acquired at the same magnification and with the fibres completely covering the focal plane.

#### 2.6.2. Quantification of DAPI, H3K4m3, PMP22, and Lamin B1 Signals and Volumes of Schwann Cell Nuclei

The 40× and 60× z-stack confocal images were analyzed using the Fiji software. We measured the fluorescence of DAPI, H3K4m3, and PMP22 in the entire z-stack of each 8-bit image of TrJ and Wt SC nuclei. In all cases, we obtained the mean intensity values (fluorescence intensity/nuclear area) provided in arbitrary units by Fiji software. To achieve quantification, we applied two methodologies using different Fiji tools on 60× confocal images.

On a set of z-stack confocal images, all Schwann cell nuclei of the sciatic nerve fiber images were delineated in each slice, using the Freehand Selections tool of the Fiji software. This procedure was carried out on the DAPI channel, and delineated nuclei represent our Objects of Interest (ROIs). To ensure the correct delimitation of the nuclei, the procedure was accomplished on the DAPI channel since, unlike the other marks, chromatin-specific DAPI staining reveals the full extent of the nucleus. Then, using the ROI manager tool of Fiji, the ROIs were transferred over the H3K4m3 and PMP22 channels, and the mean intensities of the PMP22, H3K4m3, and DAPI signals of both Wt, and TrJ were measured on the corresponding channel, plane by plane.

On the same set of 8-bit confocal images, the intensities of DAPI, H3K4m3, and PMP22 signals were also obtained utilizing the 3D manager (RoiManager3D 3.96) of the Fiji 3D plugins. First, to allow Schwann nuclei segmentation, a Gaussian blur filter (sigma: 5) was applied on duplicated images of the DAPI channel. On these duplicated images, masks of nuclei were obtained using the 3D segmentation tool of the 3D manager, establishing a low threshold (30) and a high threshold (255). Subsequently, using the Add Image tool, all the nuclear masks were uploaded to the 3D manager, and, with the Select All tool, all the nuclear masks were transferred to the DAPI, H3K4m3, or PMP22 channel raw images to measure fluorescence intensities. The 3D plugins measured plane by plane the mean intensity, considering the entire nuclear volume.

Applying both methodologies, the volume (µm3) of nuclei of Wt, and TrJ SC was measured in the DAPI channel.

Since the results obtained with both methodologies were compatible (Shapiro–Wilk and the Levene test, for normality and homoscedasticity, respectively, and Mann–Whitney tests for differences between them, α ≤ 0.05), the intensities of DAPI, PMP22, H3K4m3, and Lamin B1 signals were subsequently measured employing the 3D manager of the Fiji 3D plugins in all sets of images

#### 2.6.3. Nuclear Distributions of PMP22, H3K4m3, DAPI, and Lamin-B1 Signals

To analyze differences in DAPI, H3K4m3, PMP22 and Lamin-B1 fluorescence intensities between the central and peripheral areas of the Wt, and TrJ SC nuclei, the Straight Lines tools of Fiji software were used to trace longitudinal and transversal vectors through the nuclei. Fluorescence intensities were measured along the position coordinates of the longitudinal and transversal vectors, using the Plot Profile 2D Fiji plugins. Employing the list of values in y-coordinate (intensities), and in x-coordinate (length) given by the plugins, the central or peripheral predominance of DAPI, H3K4m3, PMP22, or Lamin-B1 intensities was established. There were obtained by dividing the x-coordinate (length) into three or four sectors, depending on whether the vector was traced transversely or longitudinally to the nucleus, respectively. The intensity values comprised in the distal two-thirds (in the case of transverse vectors) or the distal two quarters (in the case of longitudinal vectors) of the x-coordinate were considered peripheral intensity values of the DAPI, H3K4m3, PMP22, or Lamin B1 nuclear signals. Similarly, the intensity values comprised in the central third (in the case of transverse vectors) or the two central quarters (in the case of longitudinal vectors), were considered central intensity values of the DAPI, H3K4m3, PMP22, or Lamin B1 nuclear signals. Besides employing the Fiji 3D Surface Plot, a graphic 3D representation was obtained.

#### 2.6.4. Colocalization Analysis Rationale

Colocalization between two fluorescent signals can be defined in terms of two components: co-occurrence (the spatial overlap between both signals), and correlation (two signals not only overlap with one another but also co-distribute in proportion to one another). The Coloc 2 plugin calculates several co-localization coefficients to evaluate the degree of colocalization by considering both aspects mentioned above.

Manders’ coefficients (M1 and M2) estimate the co-occurrence fraction of a fluorescent signal on one channel (i.e., red color) with a fluorescent signal of another channel (i.e., green color). In other terms, these split coefficients express the fraction of a probe that overlaps another probe. Manders’ coefficients differentiate the cases when, for example, red may overlap significantly with green (M1), but most of the green may not overlap with the red (M2). These coefficients avoid issues relating to absolute intensities of the signal since they are normalized against total pixel intensity:$$M1=\frac{red\;\cap \;green}{green}\;\mathrm{and}\;M2=\frac{green\;\cap \;red}{red}$$

Manders’ coefficients can vary from 0 to 1. The former corresponds to non-overlapping images and the latter reflects a complete co-localization between both images [52,53].

Pearson’s (r), Spearman’s (*ρ*, *rho*), or Kendall’s (*τb*, *tau*) coefficients evaluate the correlation between two signals. They measure the relationship between the signal intensities in one image (i.e., channel red) and the corresponding values in another (i.e., channel green), but not the degree to which both signals co-occur. The correlation coefficient can range in value from −1 to 1. When the largest the value of the coefficient, the stronger the relationship between the variables. The sign indicates the direction of the relationship. If both variables tend to increase or decrease together, the coefficients are positive. If one variable tends to increase as the other decreases, the coefficients are negative [54]. Pearson’s coefficient (r), but not Spearman’s rank (*rho*) or Kendall’s *tau-b tau*) require normal distribution of the variables.

*Costes randomization test.* The possibility that the values obtained from the colocalization coefficients are generated by chance must be ruled out. Coloc 2 plugins include a significance test (Costes randomization test) to evaluate the probability that the value of a coefficient measured between two color channels is significantly greater than random values [55]. It is an approach by which a random probability distribution is generated by randomly scrambling blocks of pixels in one image, measuring later the correlation of this image with the other unscrambled. The random probability distribution is derived from repeated measurements of colocalization between pairs of scrambled and unscrambled images. The Costes *p*-Values should be greater than 95%.

#### 2.6.5. Colocalization Analysis of PMP22, with eu-, Heterochromatin and Lamin B1

The colocalization analysis of PMP22 with EC or HC or Lamin B1 was performed on SC nuclei employing the Coloc 2 plugins of Fiji software. Colocalization was examined on all slices of z-stack images of Wt and TrJ SC nuclei.

First, we analyzed the chromatin context in Wt and TrJ SC nuclei first, evaluating the relationships of (a) the EC with the total chromatin (TC) on cropped images with H3K4m3 or DAPI (DAPI-light/DAPI-dark) signals, respectively, and (b) the EC with the HC, employing the correspondingly mask images of H3K4m3 signal of DAPI-dark signal. Afterward, in the same Wt and TrJ SC nuclei, we analyzed the relationships of (c) PMP22 with EC, using cropped images with PMP22 or H3K4m3 signals, and (d) PMP22 with HC on respectively mask images (mask of PMP22 signal) or (mask of DAPI-dark signal).

To analyze differences between DAPI-light and DAPI-dark regions, we define a signal intensity threshold in DAPI channel, by visual inspection on each nucleus image (resolution: 1024 × 1024 pixels, pixel size 0.1 × 0.1 microns), using Fiji software. Thus, we established the pixel intensity value threshold to categorize the EC as DAPI-light and the HC as a DAPI-dark. The threshold was set at <150 intensity arbitrary units for assignment of DAPI-light euchromatic pixels and ≥150 intensity arbitrary units for DAPI-dark heterochromatic pixels. The range of intensity fluorescence belonging to DAPI-light was between ≥50 to <150, and to DAPI-dark, between ≥150 to 255.

On the DAPI channel of 8 bits merge images, each SC nucleus and the discrete surrounding region was cropped and saved separately. On the cropped images, each SC nucleus was delineated using the ROI manager tool, obtaining the nuclear ROI. To delimit the nuclear areas on H3K4m3 or PMP22 channels, the ROIs were transferred over the H3K4m3 and PMP22 channels of the same image.

Moreover, in contrast to DAPI and H3K4m3, which are specific nuclear marks, the myelin PMP22 protein extends widely in the cytoplasm. Given that, the analysis of the nuclear location of PMP22 concerning DAPI and H3K4m3 marks required the elimination of PMP22 fluorescence outside the nuclei. This was achieved using the Clear outside tool of Fiji, which deleted signals outside the nuclear ROIs. Thereby, only PMP22 nuclear marks were kept for their analysis concerning DAPI, H3K4m3, or Lamin B1marks. Additionally, masks areas of DAPI-dark (HC), H3K4m3 (EC) and PMP22 signals were obtained on the cropped images of Wt and TrJ SC nuclei, using the Threshold tool of ImageJ.

Using Coloc 2 plugin, *M1* and M2 Manders’ co-occurrence coefficients, and Spearman’s rank (*rho*) and Kendall’s (*tau*) correlation coefficients were obtained on all slices of the z-stacks cropped images of Wt and TrJ SC nuclei. Since the intensity values of PMP22, H3K4m3, DAPI or Lamin B1 signals did not fit normal distribution (Shapiro–Wilk test), Pearson’s coefficient of the Coloc 2 plugin, cannot be applied in our analysis. In addition, Costes *p*-Values greater than 95% were considered in all cases.

Using Manders’ coefficients (*M1* and *M2*) we calculated in Wt and TrJ SC nuclei:(i)*M1*: The fraction of EC (H3K4me3 signal) in the TC area (DAPI-light + DAPI-dark signals), and *M2*: the fraction of TC area (DAPI signal) in EC (H3K4me3 signal).
$$M1=\frac{\mathrm{DAPI}\;\cap \;H3K4m3}{H3K4m3}\;\;\;\;M2=\frac{H3K4m3\;\cap \;\mathrm{DAPI}}{DAPI}$$

(ii)*M1*: The fraction of EC (H3K4me3 mask) in HC (DAPI-dark mask), and *M2*: the fraction of HC (DAPI-dark mask) in EC (H3K4me3 mask).


(1)
$$M1=\frac{\mathrm{DAPI}-\mathrm{dark}\;\cap \;H3K4m3}{H3K4m3}\;\;\;\;\;M2=\frac{H3K4m3\;\cap \;\mathrm{DAPI}-\mathrm{dark}}{DAPI-dark}$$


(iii)*M1*: The fraction of EC (H3K4me3 signal) in PMP22 area (PMP22 signal), and *M2*: the fraction of PMP22 area (PMP22 signal) in EC (H3K4me3 signal).


(2)
$$M1=\frac{\;PMP22\;\cap \;H3K4m3}{H3K4m3}\;\;\;\;\;M2=\;\frac{\;H3K4m3\;\cap \;PMP22}{PMP22}$$


(iv)*M1*: The fraction of HC (DAPI-dark mask) in PMP22 area (PMP22 mask), and *M2*: the fraction of PMP22 area (PMP22 mask) in HC (DAPI-dark mask).


(3)
$$M1=\frac{\mathrm{PMP}22\;\cap \;\mathrm{DAPI}-\mathrm{dark}}{\mathrm{DAPI}-\mathrm{dark}}\;\;\;\;\;M2=\;\frac{\mathrm{DAPI}-\mathrm{dark}\;\cap \;\mathrm{PMP}22}{\mathrm{PMP}22}\;$$


(v)*M1*: The fraction of Lamin-B1 area (Lamin-B1 signal) in PMP22 area (PMP22 signal), and *M2*: the fraction of PMP22 area (PMP22 signal) in Lamin-B1 area (Lamin-B1 signal).


(4)
$$M1=\frac{\;\;PMP22\;\cap \;lamin-B1}{lamin-B1}\;\;\;\;\;M2=\frac{lamin-B1\;\cap \;PMP22}{PMP22}$$


By means of Spearman’s (*rho*) and Kendall’s (*tau*) coefficients, we analyzed the correlation between:(i)EC (H3K4m3 signal) and TC (DAPI-light + DAPI-dark signals)(ii)EC (H3K4me3 mask) and HC (DAPI-dark mask)(iii)PMP22 (PMP22 signal or mask) and EC (H3K4me3 signal or mask)(iv)PMP22 (PMP22 signal or mask) and HC (DAPI-dark mask)(v)PMP22 (PMP22 signal) and Lamin-B1 (Lamin-B1 signal).

Table 1 presents the values of the different colocalization coefficients and their meanings with respect to the colocalization results.

### 2.7. Statistical Analysis

To analyze differences between TrJ and Wt sciatic nerves, using the GraphPad Prism 8^®^ software (GraphPad Prism, RRID:SCR_002798), the Shapiro–Wilk was applied to test normal distributions concerning: (1) the number of sciatic fiber, the SC nuclei per fiber (2) the SC nuclear volumes, (3) the PMP22, DAPI, H3K4m3 and Lamin B1 mean intensities values, (4) the peripheral versus central nuclear distribution of PMP22, H3K4m3, DAPI, and Lamin-B1, and (5) the colocalization coefficients (correlations and co-occurrence coefficients) of the colocalization analysis in SC nuclei. Like none of them fit normal distributions, they were described employing the medians and 95% confidence intervals as summary measures. Accordingly, differences between Wt and TrJ were tested using the Mann–Whitney U signed-rank test (two-tailed distribution, 95% confidence, α ≤ 0.05).

## 3. Results

In the present approach, the immunostaining technique was applied to reveal the PMP22 protein in the SC nuclei of sciatic nerves of the wild (Wt) and mutated (TrJ) mice genotypes.

Both anti-PMP22 antibodies employed in the present approach (ab61220 and sc65739) can recognize both the mutated and non-mutated protein, as they match regions of PMP22 that do not include the mutation. Likewise, epitope retrieval with 70% formic acid treatment mainly removes the non-aggregated content within which a high percentage of non-mutated PMP22 is found, allowing the aggregated components, which mainly contain the mutated PMP22 protein, to persist. Therefore, the unmasked treatment mainly revealed the aggregated protein PMP22 [56].

Since the protein was remarkably located within the SC nuclei of both genotypes, the distribution of PMP22 concerning its central versus peripheral nuclear location and its colocalization with EC, HC, or lamina (Lamin B1) were analyzed.

The DAPI intensity is greater at the AT-reach and compact HC (DAPI-dark). Conversely, DAPI fluoresces with lesser intensity at GC-reach and less condensed EC (DAPI-light). Furthermore, EC can be revealed using antibodies that recognize the epigenetic modification consisting in the trimethylation of the fourth lysine residue of the histone H3 core (H3K4m3). Therefore, both H3K4m3 and DAPI-light regions delineate EC. In contrast, DAPI-dark regions without H3K4m3 signals demarcate HC.

### 3.1. Wt and TrJ Differences between Nerve Fibers and Schwann Cell Nuclei

Confocal images of Wt and TrJ sciatic nerve fibers immunostained with anti-PMP22 after epitope retrieval with 70% formic acid treatment are presented Figure 1A. The Figure also shows SC nuclei DAPI counterstained and the merge images. The distributions of the number of fibers (Figure 1B) and the number of SC nuclei (Figure 1C) of Wt and TrJ sciatic nerves were significantly higher in TrJ than in Wt sciatic nerves (*p*-Values < 0.0001). In turn, the SC nuclear volume, measured on DAPI channel (Figure 1D) was significantly higher in Wt than TrJ mice (*p*-Values < 0.0001).

### 3.2. Distinctive Intensity and Nuclear Distributions of DAPI, H3K4m3, and PMP22 Signals in Wt and TrJ Schwann Cell

Figure 2A reveals in DAPI counterstaining cropped images of both Wt and TrJ SC nuclei, the locations of the EC (H3K4m3 and DAPI-light signals), the HC (DAPI-dark signal) and the nuclear PMP22 protein (PMP22 signal). In addition to the classic localization of the myelin protein PMP22, it is interesting to note the presence of PMP22 signals within the SC nucleus, both in wild-type and mutated genotypes.

Regarding the intensities of the DAPI and H3K4m3 signals, significant differences were observed between the wild-type and mutated genotypes, with the DAPI intensity being higher and the H3K4m3 intensity lower in the TrJ nuclei than in the Wt SC nuclei (*p*-Value < 0.0001) (Figure 2B,C). Concerning the nuclear intensity of the PMP22 signal, it was significantly higher in the SC nuclei of TrJ compared to Wt (*p*-Value < 0.0001). Therefore, the nuclei of TrJ SC were smaller than those of Wt (Figure 1D); they had more PMP22 and probably more condensed chromatin (Figure 2A–C).

### 3.3. Peripheral Prevailing Location of PMP22 Signals in Wt and TrJ Schwann Cell Nuclei

The central versus the peripheral location of PMP22 in SC nuclei of Wt and TrJ mice was assessed, since visual inspection of SC nuclei suggested that PMP22 tends to be strengthened in peripheral nuclear regions (Figure 2A).

Intensity values of DAPI, H3K4m3, and PMP22 signals in relation to peripheral (P) versus central (C) distributions in Wt and TrJ SC nuclei are presented by 3D surface plots (Figure 1A) and box plots (Figure 3B,D). DAPI, H3K4m3, and PMP22 signals exhibited heterogeneous distributions across Wt and TrJ SC nuclei. The mean intensities of both chromatin labels: DAPI (Figure 3B) and H3K4m3 (Figure 3C), significantly predominated in the central regions of the Wt and TrJ SC nuclei. In contrast, PMP22 nuclear signal intensity (Figure 3D) prevailed in peripheral regions, both in the Wt and TrJ genotypes (*p*-Values < 0.0001, all comparisons).

### 3.4. Peripheral Location of PMP22 and Lamin B1 and Distinctive Intensity of Lamin B1 in Nuclei of Wt and TrJ Schwann Cell

To delineate SC nuclei and analyze the relationship between PMP22 distribution and the nuclear lamina, we immunodetected the Lamin B1 component of the lamina (Figure 4A,B) and analyzed its intensity (Figure 4C) and peripheral (P) versus central (C) distributions (Figure 4D) in the Wt and TrJ genotypes.

Lamin B1 intensity (Figure 4C) was significantly lower in TrJ than in Wt SC nuclei (*p*-Value < 0.001) and largely predominated in the SC nuclear periphery in both Wt and TrJ genotypes (*p*-Value < 0.0001), where the PMP22 signal also prevailed (*p*-Value < 0.0001). The 3D surface plots (Figure 4A), obtained on the Wt and TrJ SC nuclei and the merged images of PMP22 and Lamin B1 (Figure 4B), exemplify the aforementioned distribution. In addition, Lamin B1 spread less intensely to the central SC nuclear regions of Wt and TrJ mice (Figure 1A,B,D).

### 3.5. Eu- and Heterochromatin of Wt and TrJ Schwann Cell Nuclei

To deepen the understanding about nuclear PMP22 function, we analyzed the relationship of PMP22 with EC and HC in SC nuclei after examining the Wt and TrJ SC chromatin context. In all analyses, the Spearman’s rank (*rho*), Kendall’s tau-b (*tau*) correlation coefficients and also the *M1*, and *M2* Manders’ co-occurrence coefficients were obtained on z-stacks of cropped images or on z-stacks of cropped image masks from Wt and TrJ SC nuclei.

Figure 5 displays the colocalization results between EC (H3K4m3 signal) and TC (DAPI-light + DAPI-dark signals) and between EC and HC, by means of the respective image masks (M H3K4m3 and M DAPI-dark). Figure 5A shows images of Wt and TrJ SC nuclei with DAPI, H3K4m3, merged DAPI and H3K4m3, or PMP22 signals, and Figure 5B illustrates the corresponding image masks of DAPI-dark signal (HC), or H3K4m3 signal (EC).

As shown in Figure 5C, the *rho* and *tau* coefficients ranged from 0.8 to 1, denoting a strong positive correlation (see Table 1) between EC (H3K4m3 signal) and the TC (DAPI-light + DAPI-dark signal), in both Wt and TrJ SC nuclei. Furthermore, in both Wt and TrJ genotypes, *M1* and *M2* ranged between 0.7 and 1, indicating the co-occurrence of EC fractions in TC (*M1*), and alternatively, the co-occurrence of TC fractions in EC (*M2*). Moreover, Figure 4D illustrates the relationships between EC (M H3K4m3) and HC (M DAPI-dark), revealing *rho* and *tau* negative coefficients and *M1* and *M2* equal to 0. These results showed that, as expected, EC and HC anticolocalized. Thus, as predictable in both Wt and SC nuclei, EC colocalized with the TC and anticolocalized with HC, being both relationships significantly stronger in Wt than TrJ SC nuclei (*p*-Values < 0.0001).

### 3.6. Relationship of PMP22 with Eu- and Heterochromatin of Wt and TrJ Schwann Cell Nuclei

After examining chromatin context, the colocalization of the nuclear PMP22 (PMP22 signal) with EC (H3K4m3 signal), and the colocalization between PMP22 (PMP22 mask images) and HC (DAPI-dark mask images) were assessed in Wt and TrJ SC nuclei (Figure 6A) by means of *rho*, *tau*, *M1* and *M2* coefficients (Figure 6B,C).

Examples of images and mask images analyzed from Wt and TrJ SC nuclei are presented in Figure 6A and Figure 5B, respectively. In Wt and TrJ SC nuclei, the *rho* and *tau* correlation coefficients (Figure 6C) ranged between 0.8 and 1, signifying (Table 1) a strong positive correlation between PMP22 and EC (H3K4m3 signal). Moreover, both Manders’ *M1* and *M2* coefficients ranged from 0.7 to 1, indicating (see Table 1) the co-occurrence of EC fractions in PMP22 area (*M1*) and the co-occurrence of PMP22 fractions in EC (*M2*). Besides, all colocalization coefficients were significantly higher in Wt relative to TrJ SC nuclei (*p*-Values < 0.0001), suggesting that PMP22 protein colocalized better with EC in Wt than in TrJ genotype.

Moreover, *rho* and *tau* correlation coefficients (Figure 6D) ranged between 0.12 and 0.17, denoting (Table 1) a weak positive correlation between PMP22 and HC in both Wt and TrJ SC nuclei, with no significant differences between genotypes. In addition, *M1* and *M2* coefficients (Figure 6D) ranged from 0.15 to 0.3, indicating (Table 1) that there were small fractions of HC in PMP22 area (*M1*) and vice versa (*M2*). Figure 6D also shows that *M1* and *M2* coefficients were significantly higher in TrJ than in Wt SC nuclei (*p*-Values < 0.0001 and <0.01, respectively). This result suggests that the fraction of PMP22 was higher in the HC of TrJ than in the HC of Wt SC nuclei.

The overall correlations and co-occurrence coefficient results suggest that PMP22 colocalized with EC in both genotypes, but more in Wt than TrJ SC nuclei. Further, there was much less PMP22 colocalization with HC in both mice; however, the colocalization between PMP22 and HC was higher in TrJ than Wt SC nuclei.

### 3.7. Relationship between PMP22 and Lamin B1 of Wt and TrJ Schwann Cell Nuclei

Using the same methodological approach, we analyzed colocalization between PMP22 and Lamin B1 (Figure 7A–C).

*Rho* and *tau* correlation coefficients (Figure 7B) were close to 1, indicating (see Table 1) positive strong correlation between PMP22 and Lamin B1 signals in Wt and Trj SC nuclei. The correlation was significantly better in Wt than TrJ SC nuclei (*rho p*-Value = 0.03; *tau p*-Value = 0.0001). Additionally, the values close to 1 of *M1* and *M2* Manders’ coefficients (Figure 7C) denoted that in Wt and TrJ SC nuclei there was co-occurrence between fractions of the PMP22 and Lamin B1 areas and vice versa. However, the fraction of the PMP22 area in Lamin B1 area (*M1*) was significantly lesser (*p*-Value = 0.002) in TrJ SC nuclei concerning those of Wt, while the fraction of Lamin B1 in PMP22 areas (*M2*) was the same in SC nuclei of both genotypes.

In summary: (a) Wt and TrJ SC nuclei of sciatic nerves contain myelin protein PMP22 and Lamin B1; (b) PMP22 level was higher and Lamin B1 level was lower in TrJ than in Wt mice; (c) the number of TrJ SC nuclei in sciatic nerves was higher, the SC volumes smaller, and the amounts of chromatin higher (probably larger amount of HC) than those of Wt; (d) in Wt and TrJ SC nuclei, PMP22 and Lamin B1 predominate in peripheral nuclear regions; (e) PMP22 colocalized with Lamin B1 and with the transcriptionally competent EC, more than with the silent HC, and there were differences between both Wt and TrJ genotypes.

## 4. Discussion

We have preliminarily reported the presence of PMP22 transcript and protein in Wt and TrJ sciatic nerve fibers [51,57]. In the present work, we described the PMP22 distribution in Wt and TrJ SC nuclei and its relationship with the active EC, silent HC and Lamin B1, a component of the nuclear lamina scaffolding structure [58].

PMP22 was present in SC nuclei of both genotypes; however, its level was higher in TrJ than Wt SC nuclei (Figure 2C). Unlike the increased expression of lamin components reported in central or peripheral neuropathies caused by laminopathies [28,29,30,31], we detected and report for the first time, a lesser expression of Lamin B1 in SC of mutated TrJ concerning Wt genotype.

Nonetheless, in a mouse model for a polyglutamine disease (Dentatorubral Pallidoluysian Atrophy, DRPLA), a reduction in the nuclear level of Lamin B1 due to its cytoplasmic accumulation and excretion, correlated with nuclear degeneration (nucleophagy) and increased cell death by autophagy (Golgi-mediated-degradation, GOMED) was described [59]. These mechanisms are associated with a persistent blockade of the canonical autophagy-lysosome pathway that was also observed in human fibroblasts from patients with the neurodegenerative disease DRPLA

Lamin B1 showed similar nuclear distribution to PMP22, both predominating at the peripheral nuclear area, with lesser intensity in central regions of both Wt and TrJ SC nuclei (Figure 3A,D and Figure 4A–C). Moreover, Lamin B1 colocalized with PMP22 (Figure 7A–C), suggesting interactions between them, not only at nuclear periphery, but also in the nucleoplasmic location of both. These interesting findings will be deepened by analyzing in the future the interaction of other lamina protein components with PMP22 and other nuclear proteins in both genotypes. The lamina components residing over the nuclei were previously described, possibly forming a network that organizes different nuclear functions, such as DNA replication [58]. It was reported that nuclear lamina contributes not only to nucleoskeletal and chromatin organizing, but also to gene regulation, tissue-specific expression, and cell differentiation [58,59,60,61,62]. Besides, it has been described that cytoplasmic protein aggregates are packaged by intermediate filaments, for example, neurofilaments in neurons or keratin and vimentin in other cell types, providing stability and specificity of union [63,64]. The reduced expression of Lamin B1 (Figure 4C) and the lesser colocalization of PMP22 in Lamin B1 areas (Figure 7B,C) of TrJ than Wt SC nuclei, suggest the possible existence of the eventual lesser interactions between PMP22 and Lamin B1 in the mutated genotype. This is a horizon in our future research since we speculate that nuclear functions that need scaffolding organization could be affected by the lesser amount of Lamin B1 and the lesser interaction with at least PMP22 protein in TrJ SC nuclei.

Furthermore, the colocalization analysis between PMP22 and chromatin areas with different transcriptional statuses (EC and HC) indicated that PMP22 colocalized with the active EC in both genotypes (Figure 6A–D) and that the colocalization was higher in Wt than TrJ SC nuclei (Figure 6C). In addition, there was much less PMP22 colocalization with HC than EC in both genotypes, but the colocalization with HC was higher in TrJ than Wt SC nuclei (Figure 6D). The prevalent association of PMP22 with the transcriptionally competent EC suggests that this protein might play a nuclear function. Since the heterozygous TrJ carries a proportion of mutated PMP22, the slightly higher colocalization of PMP22 with the silent HC of TrJ than Wt SC could be associated with the mutation.

The *pmp22* were previously proposed as a growth arrest-specific (*gas3*) gene [5]. In SC cultures, transfected with retroviral vectors that constitutively over-express (sense orientation) or under-express (antisense orientation) PMP22 mRNA and protein, a cell cycle PMP22 modulation was reported. PMP22 over-expression decreased both DNA synthesis and the proportion of cell population in S + G2/M phases (flow cytometry), whereas under-expression, increased both [47]. Moreover, increased expression of cell cycle control, DNA replication and E2F genes, were detected in the Pmp22^−/−^ mutant at the onset of peripheral nerve myelination [65]. Since transgenic mice carrying additional copies of *pmp22* displayed severe hypomyelination neuropathy, expressed embryonic SC markers, and incorporated BrdU, impaired SC differentiation into the myelinating status has been also proposed [66].

We detected a greater number of SC nuclei in sciatic nerves of TrJ than Wt (Figure 1C), and preliminary data (not shown), obtained in our laboratory in three TrJ and three Wt mice, indicating BrdU incorporation into SC nuclei. This last preliminary result, which will require to be corroborated in more animals, suggests a possible SC proliferation in adult mice that seems could be higher in SC from TrJ compared to Wt. Increased proliferation of SC has been previously suggested in TrJ mice [67,68].

We speculate that PMP22 could play a role in arresting the cell cycle in G0 [5] enabling the maintenance of the SC differentiated state to allow myelination [44,45,46]. The failed PMP22 function in the hypomyelinated TrJ SC could lead to a possible higher proliferation of SC that would interfere with SC differentiation and sciatic nerve myelination. Consequently, TrJ SC could induce a compensatory but inefficient mechanism of greater transcription, as we previously reported [51], synthesis and nuclear incorporation of PMP22, explaining our described higher level of PMP22 in TrJ SC nuclei compared to Wt (Figure 2C). Further research is needed to directly demonstrate this aforementioned succession of pathophysiological events.

Hence, we agree with the proposal that PMP22 would assist at least two different functions, one related to myelination, and another to cell cycle arrest [5,69]. The CD25 transcript regulated by the P1A promoter (homologous to gene promoters with tissue-specific expression) might be mainly related to myelination. On the other hand, the ubiquitous SR13 transcript regulated by the P1B promoter (homologous to promoters of ubiquitous housekeeping genes) might be linked to cell cycle control [12,13,14,15,16,67,70]. However, the expression of both PMP22 transcripts would be necessary for myelination because myelination requires SC differentiation (SC in G0) [44,45,46].

A rising number of reports describe cellular proteins with dual functions. Some proteins, which were initially considered specific proteins of the cytoplasm or the membrane compartments, were later detected inside the nucleus, accomplishing functions of transcriptional co-activators or co-repressors. An example is the β-catenin, an integral component of adhesions junctions, which, like PMP22, belongs to the family of tight-junction claudins. β-catenin is involved in both cadherin-mediated cell adhesion and Wnt signal transduction pathway. The β-catenins of the cytoplasmic pool, can reach the nucleus and bind to the Tef/Lef-1 (an HMG box transcription factor), promoting gene expression [71,72]. Another precedent is the cytoskeletal protein, keratin 17 (K17), which was identified inside the nucleus of tumor epithelial cells, promoting gene expression and cell proliferation [73].

But, how do PMP22 and the aforementioned proteins reach the nucleus? The nuclear transport includes complex and not completely known mechanisms. The transport operates through the nuclear pore complex (NPC) composed of macromolecular arrangements of multiple copies (FG-repeat) of nucleoporins (Nup) proteins forming cylindrical channels [74,75]. Ions and molecules smaller than 40 kDa can passively diffuse through the NPC, but larger proteins require the mediation of carrier or nuclear transport factors by a facilitated carrier-mediated transport [76,77]. However, the size limit of the diffusion barrier is not rigid. For instance, by exposing hydrophobic surface residues, large proteins can cross the diffusion barrier passively [78,79].

Nup proteins also serve as specific binding sites for nuclear transport factors. The β-karyopherin (β-Kap) family, also called importins and exportins, transports most large molecules. Cargo proteins generally contain nuclear localization signals (NLS) or nuclear export signals (NES) to direct cargo in or out of the nucleus, respectively. The karyopherin-β1 (Impβ1) recognizes localization signals, but karyopherin-α (Impα), require adaptor proteins [80,81,82].

Besides, some novel transport mechanisms have been identified such as the β-catenin nuclear import through direct binding to the Nup components of the NPC [73,83]. β-catenin is a 90 kDa protein without the NLS or NES signals, the armadillo repeats (10–12) and the hydrophobic “patches” on the N and C tails of this protein associate with the Nup FG repeat facilitating the nuclear transport [83].

Furthermore, nuclear transport utilizing transporters other than importins or exportins were also described. For example, the SRY/SOX-9 components of the sex-determination pathway of mammals move into the nucleus mediated by the calcium-binding protein calmodulin, an alternative pathway to Impβ1 [84,85].

How PMP22 without NLS or NES signals pass through the nuclear envelope represents a challenger to be unveiled. Since PMP22 is a small (22 kD), highly hydrophobic protein, we speculate that individual molecules could passively diffuse one by one through the nuclear pore channel into the nucleus and then, like in the cytoplasm, would aggregate inside. However, the association of PMP22 with NLS carrying proteins or a non-classical transport mechanism (like β-catenin) might be considered and researched.

## 5. Conclusions

Taken together, our results indicate that sciatic nerves of TrJ mice, a model for human CMT1E disease, contain higher numbers of SC nuclei than their Wt counterparts. Myelin protein PMP22 was found in SC nuclei of both genotypes, with higher levels in TrJ than in Wt mice. The inverse was observed for the nuclear lamina protein Lamin B1; its levels were lower in TrJ SC nuclei than in Wt. Analysis of the distribution of Lamin B1 and PMP22 proteins showed a clear colocalization in SC nuclei. However, compared to Wt nuclei, the colocalization of PMP22 positive with Lamin B1 positive areas was significantly smaller in TrJ nuclei, suggesting that PMP22 interactions with Lamin B1 could be affected in TrJ mice. 

We further demonstrate that the nuclear distribution of PMP22 preferentially colocalized with the transcriptionally active EC, the degree of colocalization being higher in Wt than in TrJ nuclei. While some colocalization of PMP22 positive areas with HC was also observed in SC of both genotypes, it was more pronounced in TrJ than in Wt. These results are in agreement with the previously proposed nuclear function of PMP22 that is likely to be defective in the TrJ genotype. We suggest that mutated PMP22 might favor SC proliferation rather than their maturation and finally, myelination of axons. The higher level of PMP22 in TrJ SC nuclei compared to Wt could then be related to a failed compensatory mechanism. Unraveling the potential link between the abnormal SC proliferation in nerves of TrJ mice and their reduced nuclear expression of Lamin B1, remains a challenge for future research.

## Figures and Tables

**Figure 1 biomolecules-12-00456-f001:**
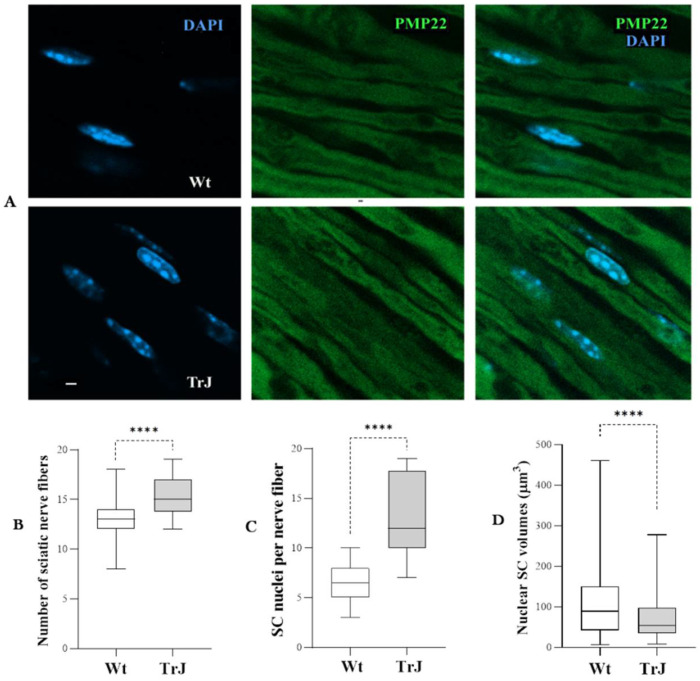
Fibers and Schwann cell nuclei in Wt and TrJ sciatic nerves. (**A**): Confocal images of Wt and TrJ sciatic nerve fibers. (**B**–**D**): Box plots illustrating the distribution of number of fibers, and SC nuclei through the whole thickness of sections and SC nuclear volumes (µm^3^) of Wt and TrJ, respectively. The fibers were immunostained using anti-PMP22 (green), after epitope retrieval (70% formic acid), and counterstained with DAPI (blue). Both areas and volumes were measured on DAPI channel. Boxes enclose the data comprising between the 25th and 75th percentiles (50% of the data). The median (50th percentile) is denoted by the transversal line within each box. Differences between Wt and TrJ were examined by means of Mann–Whitney tests (**** *p*-Values < 0.0001). Number of panoramic images analyzed = 20 per genotype. Volumes were calculated in 150 SC nuclei per genotype. Scale bar = 2 µm. Both the fiber and SC nucleus numbers were significantly larger in TrJ than in Wt sciatic nerves, whereas SC nuclear volume was significantly smaller in TrJ than Wt mice.

**Figure 2 biomolecules-12-00456-f002:**
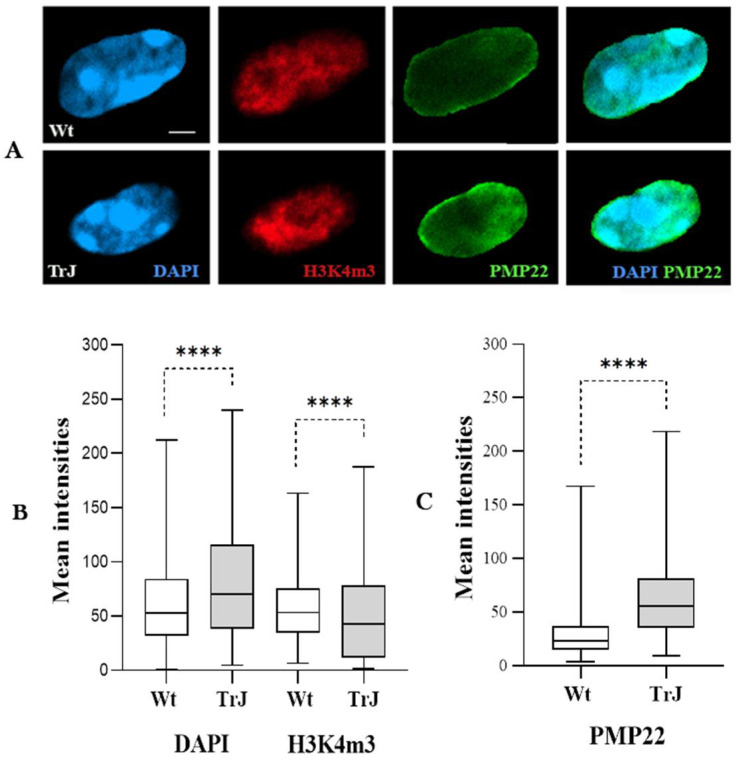
DAPI, H3K4m3, and PMP22 signals in Wt and TrJ Schwann cell nuclei. (**A**): Confocal images of a central slice of a TrJ or Wt SC nucleus. The fibers were immunostained with anti-H3K4m3 (red) anti-PMP22 (green) antibodies, after epitope retrieval (70% formic acid), and counterstained with DAPI (blue). Box plots representing the mean intensity values (arbitrary units) of DAPI and H3K4m3 (**B**) and PMP22 (**C**) signals measured on TrJ and Wt SC nuclei. The boxes containing 50% of the data, presented the median as a transversal line within them. Differences between Wt and TrJ were verified by applying Mann–Whitney tests (**** *p*-Values < 0.0001) in 150 SC nuclei per genotype. Scale bar = 2 µm. Both Wt and TrJ SC nuclei express PMP22. The intensity of PMP22 was significantly higher in TrJ than in Wt. Furthermore, the overall DAPI intensity of chromatin was significantly higher in TrJ relative to Wt, but the H3K4m3 euchromatic signal intensity was significantly lower in TrJ than in Wt SC nuclei.

**Figure 3 biomolecules-12-00456-f003:**
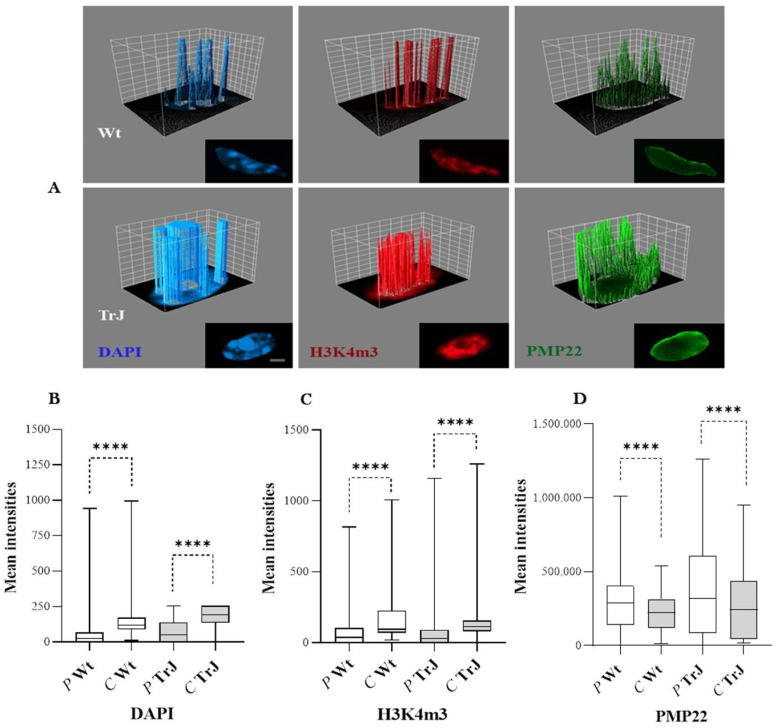
Peripheral versus central distributions of DAPI, H3K4m3, and PMP22 signals in Wt and TrJ Schwann cell nuclei. (**A**): 3D surface plot illustrating intensity (arbitrary units) of DAPI, H3K4m3, and PMP22 signal distributions upon a central plane of each Wt or TrJ SC nucleus. The peaks correspond to nuclear regions of greatest intensity (arbitrary units). The confocal images used to perform the 3D surface plots are presented below each corresponding 3D plot. Box plots show the peripheral (*P*) versus central (*C*) distribution of (**B**): DAPI, (**C**): H3K4m3 and (**D**): PMP22 signals in Wt and TrJ SC nuclei. The boxes enclose 50% of the data and represent the median and 25% and 75% quartiles. Differences between peripheral versus central distribution in Wt and TrJ were verified by applying *Mann–Whitney tests* (**** *p*-Values < 0.0001) in 100 SC nuclei per genotype. Scale bar = 2 µm. The intensities of each DAPI and H3K4m3 chromatin mark significantly prevail in the central SC nuclear regions, while PMP22 signals significantly predominate in the peripheral SC nuclear areas of the Wt and TrJ genotypes.

**Figure 4 biomolecules-12-00456-f004:**
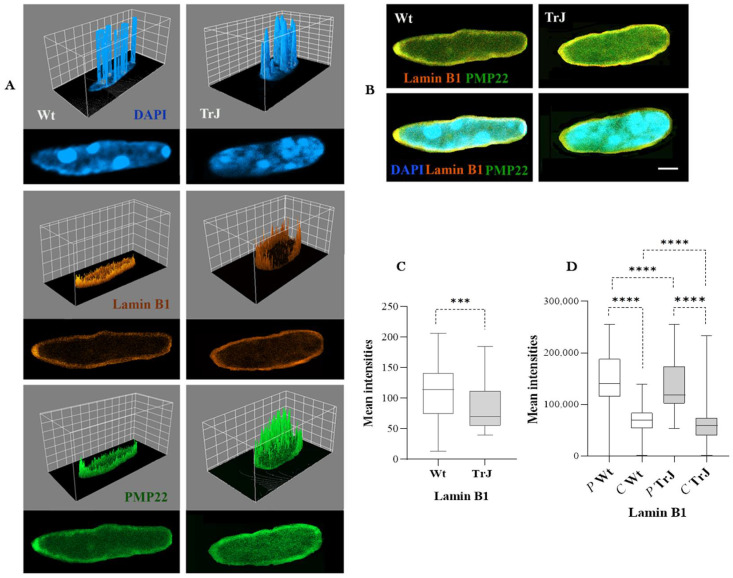
Lamin B1 intensity, peripheral versus central Lamin B1 distribution, and its relationship to PMP22 in Wt and TrJ Schwann cell nuclei. (**A**): 3D surface plot illustrating the intensity of DAPI, Lamin B1, and PMP22 signal (arbitrary units) distributions upon a central plane of each Wt or TrJ SC nucleus. The peaks correspond to nuclear regions of the greatest intensity. The confocal images of Wt and TrJ Sc nuclei are presented below each corresponding 3D plot. (**B**): Merge images of Lamin B1 and PMP22 or DAPI, Lamin B1 and PMP22. (**C**,**D**): Box plot showing Lamin B1 intensities and its peripheral (P) versus central (C) distribution in Wt and TrJ SC nuclei, respectively. Median and 25% and 75% quartiles are represented in each box. Differences between peripheral versus central distribution in Wt and TrJ were verified by applying Mann–Whitney tests (*p*-Values: *** < 0.001, **** < 0.0001) in 100 SC nuclei per analysis and genotype. Scale bar = 2 µm. Lamin B1 fluoresces less intensely in TrJ than Wt SC nuclei and preferentially localizes to peripheral SC nuclear regions where PMP22 intensity is significantly prevalent in both Wt and TrJ mice.

**Figure 5 biomolecules-12-00456-f005:**
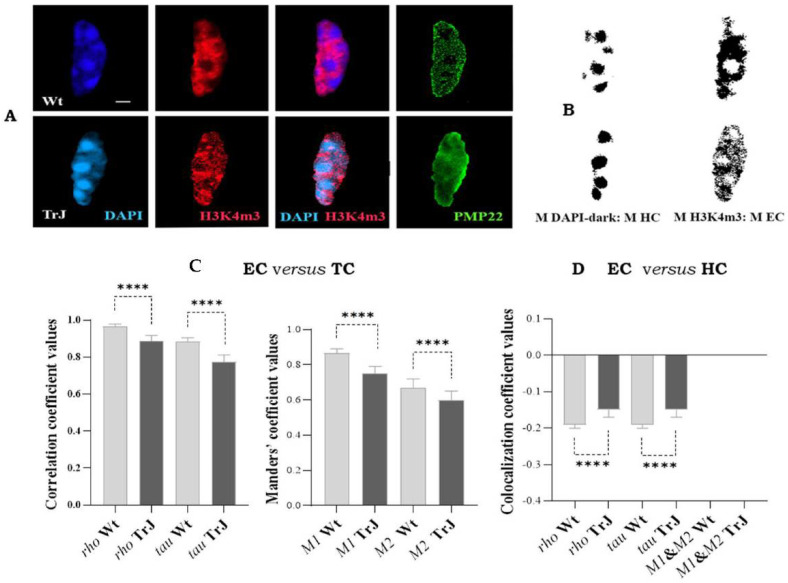
Colocalization analysis of euchromatin and heterochromatin in Schwann cell nuclei. (**A**): Images of Wt and TrJ SC nuclei showing DAPI (blue), euchromtic H3K4m3 (red) and PMP22 (green) signals. (**B**): Respectively mask images of heterocromatin (M HC) and euchromatin (M EC), obtained using the threshold tool of Fiji. (**C**): Colocalization between euchromatin (EC) and total chromatin (TC). *M1*: fraction of EC (H3K4m3 signal) in TC (DAPI-light + DAPI-dark signals), and *M2*: fraction of TC in EC. (**D**): Colocalization between EC (M H3K4m3) and HC (M DAPI-dark). *M1*: fraction of EC (M H3K4m3) in HC (M DAPI−dark), and respectively *M2*: fraction of HC (M DAPI-dark) in EC (M H3K4m3). *Rho* and *tau* correlations and *M1* and *M2* co-occurrence coefficients were estimated employing Fiji Coloc 2 plugins with Costes *p*-Values > 95%. The bar diagrams illustrate the medians and 95% confidence intervals of the coefficients. Scale bar = 2 µm. Differences between Wt and TrJ were established applying Mann–Whitney tests (**** *p*-Values < 0.0001) in 100 SC nuclei per genotype. As expected, in both Wt and SC nuclei, EC colocalizes with TC and anticolocalizes with HC, both relationships being significantly stronger in Wt than TrJ SC nuclei.

**Figure 6 biomolecules-12-00456-f006:**
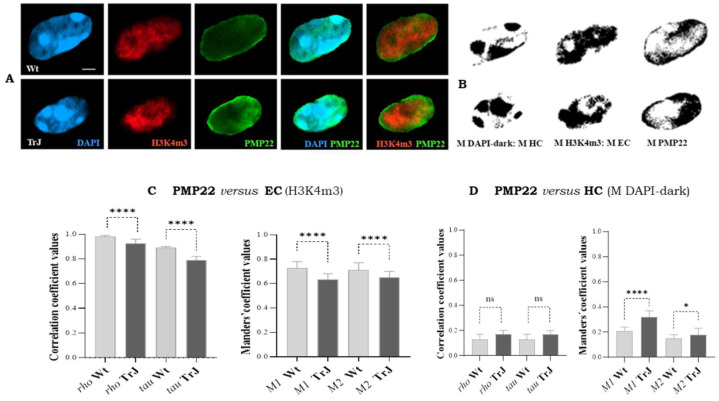
Colocalization analysis of PMP22 with euchromatin and heterochromatin in Schwann cell nuclei. (**A**): Images of Wt and TrJ SC nuclei showing DAPI (blue), euchromtic H3K4m3 (red) and PMP22 (green) signals. (**B**): Respectively mask images of heterochromatin (M HC), euchromatin (M EC) and PMP22 (M PMP22, obtained using the threshold tool of Fiji. (**C**): Colocalization analysis between PMP22 (PMP22 signal) and euchromatin (H3K4m3 signal). *M1:* fraction of euchromatin in PMP22 area, and *M2*: fraction of PMP22 area in euchromatin. (**D**): Colocalization analysis between PMP22 (PMP22 signal) and heterochromatin (M HC). *M1*: fraction of heterochromatin in PMP22 area, and *M2:* fraction of PMP22 area in heterochromatin. (**B**,**C**): Bar diagrams representing, by means of median and 95% confidence intervals, the distributions of *rho* and *tau* correlations and *M1* and *M2* co-occurrence coefficients, respectively. *R**ho*, *tau*, *M1* and *M2* were obtained employing *Fiji* Coloc 2 plugins with Costes *p*-Values > 95%. The bar diagrams represent the medians and 95% confidence intervals of the coefficients. Differences between Wt and TrJ were analyzed applying Mann–Whitney tests (*p*-Values: * < 0.01 and **** < 0.0001) in 100 SC nuclei per genotype. Scale bar = 2 µm. Correlations and co-occurrence coefficient values denote that PMP22 colocalize with EC and much lesser with HC in both Wt and TrJ SC nuclei. However, there is more colocalization with EC in Wt and more colocalization with HC in TrJ.

**Figure 7 biomolecules-12-00456-f007:**
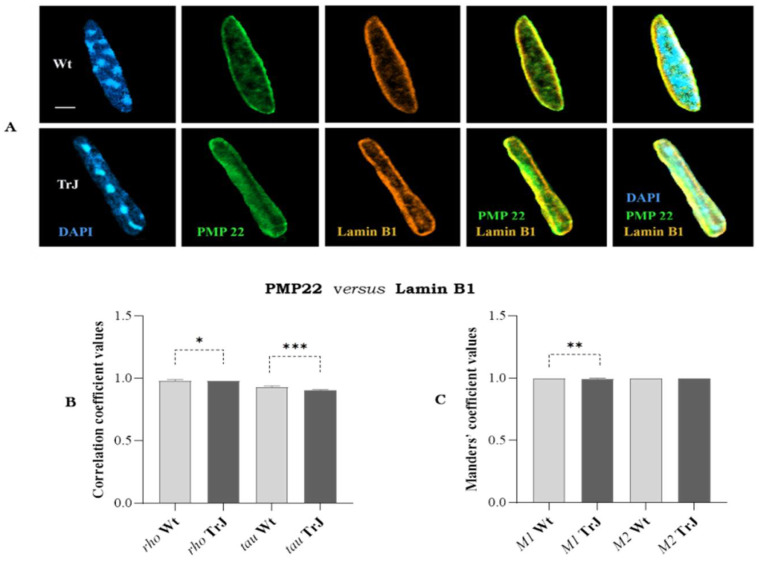
Analysis of colocalization between PMP22 and Lamin B1 in Schwann cell nuclei. (**A**): Images of Wt and TrJ SC nuclei showing DAPI (blue), Lamin B1 (brown) and PMP22 (green) signals (**B**): Distribution *of rho* and *tau* correlation coefficients between PMP22 and Lamin B1 signals. (**C**): Distribution of *M1* (fraction of PMP22 area in Lamin B1 area) and *M2* (fraction of Lamin B1 area in PMP22 22) co-occurrence coefficients. *Rho*, *tau*, *M1 and M2* were estimated employing *Fiji* Coloc 2 plugins with Costes *p*-Values > 95%. The respectively coefficient distributions are represented by bar diagrams illustrating the medians and 95% confidence intervals. Differences between Wt and TrJ were analyzed employing Mann–Whitney tests (* *p*-Values = 0.03, ** = 0.002, and *** = 0.0001 in 100 SC nuclei per genotype. Scale bar = 2 µm. Correlations and co-occurrence coefficient values indicate that PMP22 colocalizes with Lamin B1 in Wt and TrJ SC nuclei. The correlation (*rho* and *tau*) was significantly better in Wt than TrJ. The fraction of PMP22 area in Lamin B1 area (*M1*) was lesser in TrJ SC nuclei, while the fraction of Lamin B1 in PMP22 areas (*M2*) was the same in SC nuclei of both genotypes.

**Table 1 biomolecules-12-00456-t001:** Colocalization coefficient, its values and meanings.

Coefficients	Values	Meanings
M1 and M2	1	Co-occurrence
	0	No co-occurrence
rho and tau	1	Positive correlation
	0	No correlation
	−1	Negative correlation
	±0.5–±1	Strong
	±0.3–±0.49	Moderate
	±0.1–±0.29	Weak

## Data Availability

The data presented in this study are available on request from the corresponding authors.

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
