# Peer review of "Colocalization Analysis of Peripheral Myelin Protein-22 and Lamin-B1 in the Schwann Cell Nuclei of Wt and TrJ Mice"

_biomolecules, 2022, doi:10.3390/biom12030456_

Round 1
Reviewer 1 Report
see attached file.

Author Response
Reviewer #1
Thank you very much for your new review of the manuscript. Below we transcribe your questions and/or suggestions followed by our answers:
The reviewer acknowledges that the authors provided an impressively detailed response letter addressing all questions and criticisms, and introduced profound changes in the manuscript, which is indeed greatly improved.
Yet, there are still some points that may be further improved, or clarified.
- Title: nothing principally wrong with it, but in my opinion it doesn't serve the authors well because it just sounds too "trivial". I would propose something like: "Colocalization analysis of peripheral myelin protein-22 and lamin-B1 in Schwann cell nuclei of wt and TrJ mice"; this to some point insinuates that there may be differences between wt and mutant Schwann cells (which is the point of this paper, no?)
We decided to pay attention of the Reviewer proposal changing the tittle of the article by those suggested
- In their response letter the authors state: " We have not performed a comparative quantification of the total number of fibers in intact sciatic nerves of Wt with respect to TrJ." So what then represents the Fig. 1B (number of sciatic nerve fibers in Wt/TrJ)?
As we explained this response letter, the data on the number of sciatic nerve fibers were obtained by analyzing images of TrJ fibers in focal planes covering equivalent areas to those analyzed in Wt. In others words, all images analyzed in order to quantify number of fibers, were Wt and TrJ focal planes completely covered by the nerve fibers which were all acquired at the same magnification. El número total de fibras fue normalizado por el área focal total analizada en cada genotipo.
In any case, we understand that a more detailed quantitative study of the number of myelinated and unmyelinated fibers in cross sections of intact Wt and TrJ sciatic nerves, embedded in epoxy resins, should be carried out to specify and describe in greater detail the differences between genotypes. This is an interesting topic because it could indicate increased neurogenesis in the PNS under pathological conditions (a higher number of fibers equals a higher number of axons, and probably also a higher number of neurons).
- Concerning the higher number of SC nuclei in the TrJ/+ mice: this is not too much of a surprise to me, since internodes are shorter in these mice, as they are for ex. in CMT1A patients (if internodes are shorter, there need to be more SC, at least if myelination still occurs). Internodes in TrJ mice: doi: 10.1002/jnr.20357in human CMT1A: https://doi.org/10.1093/brain/awp274. Meanwhile I stepped over a publication from 1997 (!) where the number of SC nuclei et number of fibers were compared between Tr(J) and wt mice (interestingly, there was age-dependency for fiber numbers since TrJ nerves may degenerate with age; so if you continue your investigations you should pay attention to the age of your mice) (Robertson AM et al., J Anatomy 190: 423-32 (1997)). In fact, you already cited another work by the same Robertson (ref. 34).
We agree with you that this is not something new, although Liu's work was done on sensory fibers, not on sensory and motor fibers as in our case. The quantification of SC fibers and nuclei, as well as the volumetric measurement of the latter, mainly represent introductory aspects that allow the reader to recognize the experimental material, on which we perform the different analyses. In addition, they also provided elements to suggest explanations for our findings (paragraph).
We present the references of Robertson et al. (1997) and (2002) in the paragraph (lines 665-669) of the manuscript [67, 68]. Robertson et al. (2002) reported that all murine mutants examined (My41, C61 C22, and TrJ) demonstrated essentially the same type of pathology: myelin deficiency, SC proliferation, and developmental delays in myelination.All mice were 5-month-old, we will consider your suggestion in further approached.
- Still a question about the "unmasking" process with formic acid: as TrJ as well as wt sciatic nerve fibers are green all over (=PMP22 positive in Fig. 1), it is not clear how the treatment would show almost exclusively the aggregated (mutant) protein. And, if this idea about the unmasking is true, then I have some doubts about the "In summary" paragraph (lines 583-84): "PMP22 expression was higher... in TrJ than in Wt nerves", as it may have been washed away in Wt nerves? Are you talking about aggregated PMP22 only? What would PMP22 stained nerves look like without this antigen retrieval process? You have done this comparison in your JP Damian et al. paper in Biomolecules April 2021.
Aggregated protein is also present in the Wt. As mentioned in the response letter to your question: "Although some therapeutic approaches for CMT based on PMP22...". PMP22 is a low stability protein that tends to form aggregates even in its un-mutated state. Then, not only can the mutated protein aggregated. In the aggregates that remain after formic acid treatment of the nerve fibers Wt and TrJ, we detected the presence of PMP22 using antibodies against this protein. Moreover, we have not yet explored the probable presence of other proteins along with PMP22 in the aggregates after formic acid treatment.
We presented the data obtained with only the unmasking process to reduce complexity and because they reveal mainly the aggregated form and also the non-aggregated PMP22 protein (both were recognized by the same anti-PMP22).
Preliminary data (upper plot) demonstrated that PMP22 signals were higher (p-values < 0.0001 in all comparisons) in TrJ nuclei than in Wt SC, regardless of the presence or absence of the unmasking process. However, the presence of a greater amount of non-aggregated PMP22 when the unmasking process was absent would not be enough to change the overall intensity of PMP22 obtained when it was present. However, these data need to be confirmed with other approaches.
.- Figure 7B, and even more 7C: here, absolute numbers are really necessary because the difference in bar height is so small that we don't see it, despite a 2- or 3-star significance (compare M1 with M2 bars in 7C).
The differences between the values of the Wt and TrJ coefficients were small in the cases of the rho and M1, but the differences were sufficient to obtain statistical significance. The M2 coefficient was not different between Wt and TrJ.
We apply the Mann-Whitney U test that compares the sum of ranks between two data sets. The results obtained using the GraphPad program are presented below:
.-Figure 7C: M1 Wt vs TrJ (** p-value = 0,002)
Table Analyzed |
M1 M2 Wt-TrJ |
Column B |
M1 TrJ |
vs. |
vs, |
Column A |
M1 Wt |
Mann Whitney test |
|
P value |
0,0017 |
Exact or approximate P value? |
Exact |
P value summary |
** |
Significantly different (P < 0.05)? |
Yes |
One- or two-tailed P value? |
Two-tailed |
Sum of ranks in column A,B |
9883 , 7883 |
Mann-Whitney U |
3418 |
Difference between medians |
|
Median of column A |
1,000, n=94 |
Median of column B |
0,9950, n=94 |
Difference: Actual |
-0,005000 |
Difference: Hodges-Lehmann |
0,000 |
-Figure 7C: M2 Wt vs TrJ (ns: p-value = 0,3374)
Table Analyzed |
M1 M2 Wt-TrJ |
Column D |
M2 TrJ |
vs. |
vs, |
Column C |
M2 Wt |
Mann Whitney test |
|
P value |
0,3374 |
Exact or approximate P value? |
Exact |
P value summary |
ns |
Significantly different (P < 0.05)? |
No |
One- or two-tailed P value? |
Two-tailed |
Sum of ranks in column C,D |
9148 , 8618 |
Mann-Whitney U |
4153 |
Difference between medians |
|
Median of column C |
1,000, n=94 |
Median of column D |
1,000, n=94 |
Difference: Actual |
0,000 |
Difference: Hodges-Lehmann |
0,000 |
- Figure 7B: rho Wt vs TtJ (* p-value = 0.03) |
|
||||||||||||||||||||||||||||||||||||||||
|
|
||||||||||||||||||||||||||||||||||||||||
|
|
||||||||||||||||||||||||||||||||||||||||
|
|
||||||||||||||||||||||||||||||||||||||||
- Figure 7B: tau Wt vs TtJ (*** p-value = 0.0001) |
|
||||||||||||||||||||||||||||||||||||||||
|
|
||||||||||||||||||||||||||||||||||||||||
|
|
||||||||||||||||||||||||||||||||||||||||
|
|
- I don't like the expression "...SC nuclei expressed PMP22...". Proteins may be present in nuclei, or genes may be expressed, but the nuclei themselves do not "express proteins". The more so since PMP22 is transported into the nucleus, even if we don't know how (Discussion).
We know and agree with the biological concept of gene expression, but in this case “expression” means the visualization of signals corresponding to the nuclear protein PMP22. However, following your suggestion, throughout the current manuscript, we use a different term to indicate the presence of PMP22 in SC nuclei, such as presence, levels.
Personally, I would start the Discussion with line 608, and not with some trivial facts about HC and EC. These have their place and are mentioned anyway from line 644 on (chromatin colocalization). The paragraph lines 600-607 may follow later, and is rather speculative.
To our knowledge, the presence of intense DAPI signals is an indirect evidence of nuclei with more condensed chromatin and probably lesser transcriptional status. However, following your suggestions we deleted the paragraph lines 600-607 and started the Discussion on line 608.
- Finally, even if the response letter says that the English was revised, it would be good to re-revise it. This is particularly true for the "5: Conclusions" section that has been entirely rewritten, and which probably every reader will study with specific attention.
To "acknowledge" your effort on the ms and the answer letter, this is my proposition for this last paragraph (you are obviously free to use it or not):
Taken together, our results indicate that sciatic nerves of TrJ mice, a model for human CMT1E disease, contain higher numbers of SC nuclei than their Wt counterparts. Myelin protein PMP22 was found in SC nuclei of both genotypes, with higher levels in TrJ than in Wt mice. The inverse was observed for the nuclear lamina protein Lamin B1, its levels were lower in TrJ SC nuclei than in Wt. Analysis of the distribution of Lamin B1 and PMP22 proteins showed a clear colocalization in SC nuclei. However, compared to Wt nuclei, the overlap of PMP22 postive with Lamin B1 positive areas was significantly smaller in TrJ nuclei, suggesting that PMP22 interactions with Lamin B1 could be affected in TrJ mice. We further demonstrate that the nuclear distribution of PMP22 preferentially coincided with the transcriptionally active EC, the degree of colocalization being higher in Wt than in TrJ nuclei. While some overlap of PMP22 positive areas with HC was also observed in SC of both genotypes, it was more pronounced in TrJ than in Wt. These results are in agreement with the previously proposed nuclear function of PMP22 that is likely to be defective in the TrJ genotype. We suggest that mutated PMP22 might favor SC proliferation rather than their maturation and finally, myelination of axons. The higher expression of PMP22 in TrJ SC nuclei compared to Wt could then be related to a failed compensatory mechanism. Unraveling the potential link between the abnormal SC proliferation in nerves of TrJ mice and their reduced nuclear expression of Lamin B1, remains a challenge for future research.
We thank the Reviewer and decided to replace our way of writing the conclusions with the suggested one. The only difference with the text you wrote is the substitution of the word “overlap” for the term “colocalize”, considering that colocalization does not mean literal overlap.

Reviewer 2 Report
I thank the authors who modified the article, following the reviewers' advice, changing the text, figures and legends. Now the paper is ready to be published.
Author Response
There is no comment to the Reviewer #2
Reviewer 3 Report
The authors showed that the sciatic nerves of TrJ mice contain more Schwann cell nuclei than Wt mice, and show that there is colocalization of PMP22 and Lamin B1 in the nuclei in both genotypes. But trJ mutants have higher PMP22 expression and lower Lamin B1 expression, which could be related to SC proliferation. The authors have improved the manuscript and addressed all reviewers' comments.
I would like to know why authors have used only male in their studies?
Female are also affected by these mutation.
Are the WT animales littermates? Some small changes in the paper could be explained if the animals are from different mouse colonies.
What is the author’s opinion on morphology of the nuclei and the SC phenotype? The mutant mice are hypomyelinated and the SC nuclei may have different morphology and chromatin regions due to the lack of myelin.
What age of mouse are authors using in their experiments? Chromatin remodeling enzymes are very different during SC development and the chromatin will be different. (https://pubmed.ncbi.nlm.nih.gov/28850819/)
How do the authors distinguish myelin (or Remak) SC from SC Supernumerary in their mutant sciatic nerves?
This supernumerary are increased in trJ mice and have a different gene expression and nuclei could be different. (https://pubmed.ncbi.nlm.nih.gov/12090404/)
(Or myelin SC from Remak SC in their WT nerves)
Minor points:
Could the authors make box plots figures for figures 5, 6 and 7?
Author Response
Reviewer #3
Thank you very much for your new review of the manuscript. Below we transcribe your questions and/or suggestions followed by our answers:
The authors showed that the sciatic nerves of TrJ mice contain more Schwann cell nuclei than Wt mice, and show that there is colocalization of PMP22 and Lamin B1 in the nuclei in both genotypes. But trJ mutants have higher PMP22 expression and lower Lamin B1 expression, which could be related to SC proliferation. The authors have improved the manuscript and addressed all reviewers' comments.
- I would like to know why authors have used only male in their studies? Female are also affected by these mutation.
Considering that there could be a possible influence of gender on the results, we decided to use only male animals to reduce the complexity of the work. It is interesting to consider the analysis of female mice in a later approach.
- Are the WT animales littermats? Some small changes in the paper could be explained if the animals are from different mouse colonies.
Each analysis was performed strictly in Wt and TrJ littermates.
- What is the author’s opinion on morphology of the nuclei and the SC phenotype? The mutant mice are hypomyelinated and the SC nuclei may have different morphology and chromatin regions due to the lack of myelin.
The distribution of the chromatin regions would be an intrinsic nuclear property, probably influenced by the global transcriptional expression. As we explained in the previous response letter, SC nuclei of the Wt genotype are generally elongated and appear parallel to the major axis of the teased sciatic nerve fiber. Under the same condition, TrJ SC nuclei have a slightly more spherical appearance. The difference in the spatial arrangements of the nerve fibers and the morphology of the SC nuclei between TrJ and Wt could be a consequence of a reduction in the packing of the TrJ nerve fiber due to the absence of collagen IV in the basal lamina (Rosso et al. al., 2012) [59]. The smaller volumes of TrJ SC nuclei compared to those of Wt may be related to a higher proportion of compact HCs in TrJ CS.
.- What age of mouse are authors using in their experiments? Chromatin remodeling enzymes are very different during SC development and the chromatin will be different. (https://pubmed.ncbi.nlm.nih.gov/28850819/)
Thanks you very much for the information. As we documented in Material and Method section, all mice were 5-month-old. Accordingly, we expect that the pool of chromatin remodeling enzymes were relatively equal in all animals.
- How do the authors distinguish myelin (or Remak) SC from SC Supernumerary in their mutant sciatic nerves? This supernumerary are increased in trJ mice and have a different gene expression and nuclei could be different. (https://pubmed.ncbi.nlm.nih.gov/12090404/) (Or myelin SC from Remak SC in their WT nerves)
Remak fibers are amyelin fibers. These fibers do not generate myelin around the axon. In these fibers, a single Schwann cell surrounds several small axons (usually less than 1 um in diameter, Figure 1, R). They differ structurally from myelin fibers, not only because of the presence of spiral turns of their membrane around the axon that forms the myelin sheath, but also because the axons are larger. This can be seen in our micrography of a human sural nerve (Figure 1, M). To identify myelinated fibers, the best structural approach is transmission electron microscopy (TEM) and possibly confocal microscopy with myelin-specific markers (ie, BASHY Fluorphore), but the level of resolution makes it impossible to clearly determine amyelinated from myelinated fibers.
Figure 1
When we compare WT with TrJ fibers by TEM, differences in Schwann cell arrangement, dimensions, and myelin formation are evident (Figure 2 A: TrJ and B: Wt, bar: 250 nm). The thinness of the myelin and the disorganization of the extracellular matrix in a myelin fiber in TrJ are striking (Figure 2A, M). It is possible to identify Remak fibers in the same image (Figure 2A, R). We have not yet characterized supernumerary Schwann cells as described by Robertson et al. (2002). However, our work is in progress and in the future we could explore a more detailed characterization of the cell populations of TrJ nerve fibers.
Figure 2
- Minor points: Could the authors make box plots figures for figures 5, 6 and 7?
Bar diagrams, illustrating medians and 95% confidence intervals, are appropriate ways to represent the distributions of variables, and were clearer than box plot to our purpose. Since graphs of Figures 5, 6 and 7 represent distribution of coefficients and not attributes or characteristics of the CS nuclei, we considered that it was not necessary to diagram their population distribution by blot plot. In addition, as the colocalization coefficients have little values and variable dispersions, it is easier to understand the differences between the Wt and TrJ genotypes if we only graph their medians. We present the box plots below and only if you consider it strictly necessary, we would change them in the manuscript.

This manuscript is a resubmission of an earlier submission. The following is a list of the peer review reports and author responses from that submission.
Round 1
Reviewer 1 Report
Solid data, well done. A minor point though, I would recommend the improvement of the resolution in the figures.
Author Response
Thank you for reviewing the manuscript. Below we transcribe your suggestion followed by our answer:
Reviewer: A minor point though, I would recommend the improvement of the resolution in the figures
Answer: The images were acquired with a Zeiss 800 confocal microscope, using a 63x objective, with a numerical aperture of 1.42, and a resolution of 1024 x 1024 pixels (pixel sizes of 0.1 x 0.1 microns µm). Under the described conditions, z-stack images were acquired every 0.21 µm. The distinct variables were quantified in each plane of each Wt and TrJ SC nuclear image which resolution was adequate to carry out all the analyses. The methodologies employed to perform them are exhaustively described in the Materials and Methods of the manuscript.
The example images that we present in the manuscript are those that were quantified to maintain agreement between the analysis and the illustration of the data.
We could present images at a higher resolution, for example at 2048 x 2048 pixels, but they would not represent what was analyzed. In this manuscript we have improved the quality of the images, using tools from the Fiji program.
Reviewer 2 Report
Di Tomaso and colleagues present a paper entitled "The PMP22 myelin protein is expressed in Schwann cell nuclei"
The authors of this article show how the myelin protein PMP22 could play a nuclear role by emphasizing a compensatory mechanism for its defense functions. The work is generally well written and well organized. The various parts are clear, and the results illustrate the data presented in the figures very well.
Only minor details could be improved and changed.
- In the text, figure 1E is indicated, but in figure 1 the panel E does not exist. Clarify this point.
- Some panels of figures 4A, 5A, 6A and B are not considered in the results. As far as possible, I recommend inserting them.
- The conclusions are well written and appropriate to the results presented. Overall, the section is probably very long. If necessary, it is advisable to shorten and reduce any repetitions present.
- References are highlighted in the text with the name/year and as numbers, as evident in the references section. The sequence of numbers also does not follow a sequential criterion. Check and standardize.
Author Response
Thank you for your careful review of the manuscript. Below we transcribe your questions and/or suggestions followed by our answers:
Reviewer: In the text, figure 1E is indicated, but in figure 1 the panel E does not exist. Clarify this point. Some panels of figures 4A, 5A, 6A, and B are not considered in the results. As far as possible, I recommend inserting them.
Answer: “Figure 1 E” was removed and substituted by “Figure 1 D”. The “A parts” of all Figures are described in the Results section of the current version of the manuscript.
Reviewer: Overall, the section is probably very long. If necessary, it is advisable to shorten and reduce any repetitions present.
Answer: We take your feedback into account and shortened the text in all sections of the new manuscript version, trying to avoid repetition.
Reviewer: References are highlighted in the text with the name/year and as numbers, as evident in the references section. The sequence of numbers also does not follow a sequential criterion. Check and standardize.
Answer: We checked, corrected, and standardized all the references included in the current manuscript version.
Reviewer 3 Report
In the present manuscript, authors researched the novel nuclear localization of normal and mutated PMP22 in SCs and CA3 neurons of Wt and TrJ mice. The distribution characteristics of PMP2, euchromatin and heterochromatin are analyzed. Furthermore, relationships of PMP22 with eu- and heterochromatin are discussed. I have some comments on the manuscript as follows.
- How to define the DAPI-light region and DAPI-dark region? Please describe the specific threshold (the range of fluorescence intensity) to distinguished the DAPI-light region and DAPI-dark region in the manuscript.
- In Figure 1A,besides the merged images, it is better to add the individual images of DAPI and PMP22 for the perspicuous exhibition of PMP22 nuclear localization.
- The main text (572-582 lines) lacks the description about results of Figure 5A and Figure 5C.
- Could authors add the analysis about colocalizations of PMP22 with eu- and heterochromatin in CA3 nuclei (rho and tau correlations and M1 and M2 Manders´ co-occurrence coefficients) to define whether the relationship of PMP22 with eu- and heterochromatin in CA3 nuclei is same as that in SC nuclei?
- Could authors co-stain the PMP22 and the nuclear envelope marker such as Lamin B to confirm the peripheral localization of PMP22 furtherly and detect whether PMP22 interacts with Lamin B.
- There are some mistakes of labels in the manuscript. For example, at 452 line, “B-C” should be changed to “B-D”. At 467 line, where is the “Figure 1E”? At 493 line, “Figure 2B” should be changed to “Figure 2B-C”. At 600 line, “Figure 5B” should be changed to “Figure 6B”. The caption of Figure 4 (at 529 line) is same as that of Figure 5 (at 559 line). It is obvious that Figure 4 showed the colocalization analysis of euchromatin and heterochromatin. In addition, the content in parenthesis of Figure 5C should be “M DAPI-dark”. Please check labels carefully.
- As we all know, protein localization is closely related to protein function, but what specific role will PMP22 verification play in this paper?
Author Response
Thank you very much for your careful review of the manuscript. Below we transcribe your questions and/or suggestions followed by our answers:
Reviewer: 1- How to define the DAPI-light region and DAPI-dark region?
Please describe the specific threshold (the range of fluorescence intensity) to distinguished the DAPI-light region and DAPI-dark region in the manuscript.
Answer: 1- To analyze differences between DAPI-light and DAPI-dark regions, we define a signal intensity threshold in DAPI channel, by visual inspection in each nucleus image (resolution: 1024 x 1024 pixels, pixel size 0.1 x 0.1 microns, 63x lens, 2x zoom), using FIJI software Thus, we established the pixel intensity value threshold to categorize as DAPI-light the EC and as a DAPI-dark the HC. The threshold was set at < 150 intensity arbitrary units for assigning DAPI-light euchromatic pixels and ≥ 150 intensity arbitrary units for DAPI-dark heterochromatic pixels. The intensity fluorescence range belonging to DAPI-light was comprised between ≥ 50 to <150, and to DAPI-dark, between ≥150 to 255. The above description is now included in the Materials and Methods section of the new manuscript.
Reviewer: 2- In Figure 1A, besides the merged images, it is better to add the individual images of DAPI and PMP22 for the perspicuous exhibition of PMP22 nuclear localization.
Answer: 2- Along with the combined Wt and TrJ sciatic nerve images, the corresponding individual DAPI and PMP22 images were entered into Figure 1A in the current manuscript version.
Reviewer: 3- The main text (572-582 lines) lacks the description about results of Figure 5A and Figure 5C.
Answer: 3- We have introduced in the corrected manuscript version the descriptions of Figures 5 A and B (Figure 6 A and B in the present corrected version) which were involuntarily omitted in the first version.
Reviewer 4- Could authors add the analysis about colocalizations of PMP22 with eu- and heterochromatin in CA3 nuclei (rho and tau correlations and M1 and M2 Manders´ co-ccurrence coefficients) to define whether the relationship of PMP22 with eu- and heterochromatin in CA3 nuclei is same as that in SC nuclei?
Answer: 4- We will perform the analysis suggested on CA3 and also include the hippocampal dentate gyrus analysis. We have decided not to include the results obtained in CA3 in this manuscript in order to complete and present them in a future manuscript.
Reviewer: 5- Could authors co-stain the PMP22 and the nuclear envelope marker such as Lamin B to confirm the peripheral localization of PMP22 furtherly and detect whether PMP22 interacts with Lamin B.
Answer: 5- The results related to this request were included in the present manuscript version. Figures 4 and 7, illustrated these results, and for this reason, the title of the manuscript has been changed.
Reviewer: 6- There are some mistakes of labels in the manuscript. For example, at 452 line, “B-C” should be changed to “B-D”. At 467 line, where is the “Figure 1E”? At 493 line, “Figure 2B” should be changed to “Figure 2B-C”. At 600 line, “Figure 5B” should be changed to “Figure 6B”. The caption of Figure 4 (at 529 line) is same as that of Figure 5 (at 559 line). It is obvious that Figure 4 showed the colocalization analysis of euchromatin and heterochromatin. In addition, the content in parenthesis of Figure 5C should be “M DAPI-dark”. Please check labels carefully.
Answer: 6- All corrections remarked by the Reviewer have been rectified; however, the corresponding line numbers are different in the new manuscript version.
The title of Figure 4 (Figure 5 in the current version) was changed as follows: “Colocalization analysis of euchromatin and heterochromatin in Schwann cell nuclei”.
The content in parenthesis of Figure 5C (Figure 6C in the present version) was replaced by “M DAPI-dark”.
Reviewer: 7- As we all know, protein localization is closely related to protein function, but what specific role will PMP22 verification play in this paper?
Answer: We especially thank the Reviewer for this question. Our thinking was based on our current data, our preliminary results, and reported findings in the scientific literature.
We summarized our finding below:
- The location of PMP22 into SC nuclei of Wt and TrJ sciatic nerves fiber
- The greater amount of PMP22 expressed in CS nuclei of TrJ than Wt
- The colocalization of PMP22 with transcriptional competent EC in both Wt and TrJ genotypes, but with more colocalization in the normal Wt
- The lesser colocalization of PMP22 with de silent HC than the EC in both genotypes, however with slightly more PMP22/HC colocalization in the mutated TrJ
- The higher number of CS nuclei observed in sciatic nerves of TrJ regarding Wt
These results integrate with others reported in the literature to shed light on the complex functions of PMP22. In this sense, we would like to highlight the following contributions:
- Our previously reported higher amount of PMP22 transcripts in SC nuclei and cytoplasm of TrJ regarding Wt [51,58]
- The previously proposed role for pmp22 as a growth arrest-specific (gas3) which is up-regulated by serum starvation (G0 cell cycle) [5]
- The negative gene modulation of SC cell proliferation by PMP22 [43]
- The requiring SC differentiation to achieve the myelination [40-42]
- Our preliminary results obtained in three TrJ and three Wt mice that showed greater incorporation of BrdU in TrJ CS nuclei than in Wt (data not shown in the present manuscript).
- The altered myelination of TrJ and CTM1E disease [31,32].
The presence of intranuclear PMP22 and its association with the transcriptionally competent chromatin context, mainly in the normal genotype, suggests the nuclear incorporation of PMP22 to fulfill a conceivable nuclear function. The role previously proposed for pmp22 as a specific growth arrest gene and as a negative modulator gene for cell proliferation suggest that PMP22 could contribute to maintaining SC in G0, inhibiting its proliferation to allow SC differentiation, which is necessary to achieve the myelination process. This function of PMP22 could be partially affected in TrJ SC nuclei. We speculate that TrJ SC nuclei carrying mutated pmp22 would fail to maintain the differentiated state, and continue to proliferate. This undifferentiated SC state would affect nerve myelination. The SC would attempt to compensate for impaired PMP22 function by transcribing, synthesizing, and incorporating more PMP22 into the SC nuclei of TrJ than Wt. New approaches will be needed to unravel the functions of this protein. Our findings open up a field of future research, which we are beginning to delve into.
We would like to point out some considerations that allowed us to contextualize our approach. PMP22 is a small, highly hydrophobic protein that, under normal conditions, has chaperone-assisted folding, with a high propensity to aggregate to form perinuclear aggresomes. More than 80% of neosynthesized PMP22 has difficulty inserting into the membrane and is rapidly degraded by the proteasome (Notterpek et al., 1999; Parek et al, 1997). Although PMP22 is a ubiquitous protein, it is in SC where it has the highest expression levels (Li et al, 2013). PMP22 profoundly affects SCs, especially due to problems related to folding, insertion, degradation, and the formation of metabolically active aggregates (Rangaraju et al, 2009). The functional cost of myelin homeostasis with correct integration of PMP22 is very high, given that only 20% of the protein synthesized under normal conditions is inserted into the membrane. This implies a high energy expenditure not only in the synthesis of the protein that will not finally be integrated into the membrane but also in the maintenance of the proteasomal machinery responsible for the clearance of the excess protein not integrated into the membrane. Why is this "expensive" protein still being synthesized? What is the “benefit” offered by their roles? The nuclear presence of PMP22 and its early characterization as GAS3 add complexity to the understanding of its possible functions, which also occurs with other claudin family proteins, such as PMP22, catenins. However, although its expression is higher in G0, the pmp22 gene is expressed in all stages of the cell cycle, indicating other possible roles, with an influence on cell homeostasis. For example, some authors suggest that PMP22 and PERP (another protein of the PMP22/GAS3 family) participate in a balance that involves regulation of the cell cycle by PMP22 and increases apoptosis by PERP, via p53 (Attardi et al., 2000; Hou et al., 2021). Cell cycle control is integrated into myelinating CS with other transcendental events such as myelination itself and inhibition of axonal sprouting and both are supported by the transcellular dialogue between glia and neurons, for the maintenance of homeostasis of the peripheral nerve fiber. PMP22 could be playing a crucial role (and in this sense, structurally and functionally "transversal") in the synchronization (alone or as a component of supramolecular, nuclear, or cellular complexes) of at least two of these important events in CS, cell arrest and myelination, to which the balance between maintenance in G0/apoptosis, or another unknown function could eventually be incorporated.
In this sense, elucidating the mechanism(s) of entry of PMP22 into the nucleus could probably lead to revealing an eventual association of PMP22 with other proteins that negatively could modulate the cell cycle
- Notterpek L, Ryan MC, Tobler AR, Shooter EM (1999). PMP22 accumulation in aggresomes: Implications for CMT1A pathology. Neurobiol Dis 6: 450–460
- Pareek S, Notterpek L, Snipes GJ, Naef R, Sossin W, Laliberté J, Iacampo S, Suter U, Shooter EM, Murphy RA (1997) Neurons Promote the Translocation of Peripheral Myelin Protein 22 into Myelin. Neurosci 17(20):7754-7762.
- Li J, Parker B, Martyn C, Natarajan C, Guo, J (2013). The PMP22 gene and its related diseases. Mol Neurobiol 47: 673–698
- Rangaraju S, Hankins D, Madorsky I, Madorsky E, Lee W-H, Carter CS, Leeuwenburgh C, Notterpek L (2009). Molecular architecture of myelinated peripheral nerves is supported by calorie restriction with aging. Aging Cell 8(2):178-191.
- Attardi LD, Reczek EE, Cosmas C, Demicco EG, McCurrach ME, Lowe SW, Jacks T (2000). PERP, an apoptosis-associated target of p53, is a novel member of the PMP-22/gas3 family. Genes Dev 14:704–718.
- Hou J, Wang L, Zhao J, Zhuo H, Cheng J, Chen X, Zheng W, Hong Z, Cai J (2021). Inhibition of protein PMP22 enhances etoposide-induced cell apoptosis by p53 signaling pathway in Gastric Cancer. Int J Biol Sci 17(12): 3145–3157.
Reviewer 4 Report
The manuscript reports on a nuclear localization of the myelin protein PMP22, mainly expressed in Schwann cells (SC). This is principally interesting both for clinical and fundamental science since pmp22 gene overexpression, deletion, or point mutations are largely responsible for different Charcot-Marie-Tooth diseases, notably CMT1A (gene duplication), HNPP (PMP22 under-expression), and CMT1E (point mutations). The authors used a PMP22-specific antibody to study the local distribution of the protein in nuclei of wildtype (wt) and Trembler-J (TrJ) mice SC (a model for CMT1E), in particular its co-localization with eu- or heterochromatin. Both mutated and wt PMP22 are essentially colocalized with euchromatin, but some mutated PMP22 is also found together with heterochromatin. Based on their observation of a higher PMP22 level in TrJ nuclei than in wt, the authors suggest a potential role for PMP22 in cell cycle arrest in G0, which may be perturbed in TrJ mice (or CMT1E patients); TrJ mice might compensate for lack of functional PMP22 by stronger expression in nuclei.
To the reviewer, the manuscript has certain shortcomings, as detailed below.
Overall, the reviewer sees like an "imbalance" between reported facts and length of the paper. Much of the ms seems more of a review, than a research article (almost 3 print pages of Introduction, and a large part of the Discussion section dealing with findings not directly related to the present work, but rather general knowledge about PMP22).
the title: "The PMP22 myelin protein is expressed in Schwann cell nuclei": the nuclear presence of PMP22 in SC is not really a new fact, as the authors themselves already described it for ex. in their publication Kun et al. 2012 (book chapter; see e.g. Fig.4 in that work), DOI: 10.5772/35306. More on this later.
The rationale for the research is contained in Introduction, lines 118-140. In my eyes, the argumentation falls a bit short; thus, current research does not consider that demyelination in CMT1E is due only to misfolding of mutated PMP22 and toxic aggregation with normal PMP22 affecting membrane transport and degradation. For example, already a slight stoichiometric imbalance between the different myelin components (PMP22 interacts with P0) is fatal for myelin formation and maintenance, among still other causes implying PMP22 protein.
The authors then argue that, to explain CMT disease onset, a potential role of PMP22 in cell cycle control has so far not been taken into account "as adjunctive cause", although the gene had originally been discovered as gas-3 (growth arrest) gene in NIH3T3 fibroblasts.
This argument is however questionable since in fact, a role for PMP22 in cell survival and/or proliferation was suggested very early, some of the corresponding literature is even cited in the ms (ex., refs. [5], [78]). In 2005, Giambonini-Brugnoli et al. (not cited) write: "Our findings [...] provide the first in vivo evidence for a direct role of PMP22 in the regulation of cell proliferation." And later: "Increased expression of genes involved in cell cycle regulation and DNA replication is characteristic and specific for the early stage in Pmp22−/− mice, supporting a primary function of PMP22 in the regulation of Schwann cell proliferation. In the Tr mutant, a distinguishing feature is the high expression of stress response genes..." (Neurobiology of Disease 18:656-68). In human SC, PMP22 was reported to function as growth arrest gene acting on multiple signaling pathways and transcription factors like Knox20, EGR2, Sox10 (see for ex. ref. [19]: Li et al. 2013, Mol Neurobiol 47(2):673-98; a very detailed review on all aspects of PMP22 and CMT). Is there a clear difference between the proposal of these former studies, and that of the present manuscript?
To me, it is not really clear what exactly the authors mean by "a defective cell cycle" due to the TrJ mutation (or whether this would differentiate their viewpoint from previous studies), and at what stage of development this should be important. Sure, myelination starts only when SC proliferation stops (G0). Thereafter, all myelin proteins have to be regularly renewed, with all of them having different lifespans. Unfortunately, elucidation of PMP22 function is indeed a complex task, due for ex. to the fact that it behaves differently between in vivo and in vitro: Thus, PMP22 overexpression reduces SC proliferation in vitro, but enhances it in vivo, the inverse being true for PMP22 deficiency. Point mutations like in TrJ suppresses SC proliferation in vitro, but increases it in vivo (the latter may be related to demyelination-induced reactivation of SC (Li et al. 2013).
Concerning the content of the ms, while the immunocytochemical part and in particular, the colocalization studies are well conducted (with quite sophisticated image and statistical analyses making up for a big part of the ms), the reviewer regrets that the theory advanced by the authors is entirely based on "simple" fluorescence staining of SC nuclei with a single antibody. This falls even behind the above-cited work of 2012 where IHS was also employed to detect pmp22 RNA in the nucleus. By the way it would have been interesting to discuss the difference between the results reported in 2012 and in the present ms: in the 2012 paper PMP22 protein had been detected only in TrJ, not in wt nuclei (different antibodies, different antigen retrieval protocols, different microscopy, artifact?).
When the authors claim "mutated PMP22 was more colocalized with heterochromatin than normal PMP22" (Abstract line 30, and elsewhere): It is unlikely that their antibody staining approach can distinguish between wt and mutated PMP22 protein. Moreover, since the TrJ mouse is heterozygous for the mutation, can we only be sure that the nuclear expression concerns indeed the mutated protein? It would have been interesting to know whether the higher presence of PMP22 in TrJ nuclei was essentially due to mutated protein, or wt, or maybe even a complex/aggregate of both? Is the mutated gene generally (not only in the nucleus) upregulated in the cell to make up for lack of normal PMP22 protein (which may have as corollary a higher transport into the nucleus)? Nuclei are rather easy to purify by centrifugation, so even Western blotting or other analysis methods could be employed.
Thus, in the end, the scientific facts on which the authors have based their theory (cell cycle regulation by PMP22, defective if mutated) appear a bit weak, based on rather indirect evidence and statistical correlation/coincidence. The authors themselves seem not sure how to interpret their data, resp. how to put them into the context of their theory; at the end of the "Conclusion" section we read: "The prevalent PMP22 location at peripheral nuclear regions of both genotypes could indicate its access and level of diffusion into the nucleus or a possible association of PMP22 with nuclear lamina or with active or inactive chromatin associated with the nuclear lamina". This seems a bit vague and it is not self-evident that these observations are indicative of a cell cycle regulation role for PMP22.
At the same time, quite many PMP22-defective mouse models are available today. The reviewer thinks that already a similar antibody staining of those mice could probably contribute to verify the authors' theory: what happens to nuclear staining when PMP22 is under- or over-expressed to different degrees; and how is this correlated to SC cell cycle?
While it is in line with the authors' theory that the number of SC nuclei is higher in TrJ mice, is there an explanation why there seem to be also more fibers in TrJ mouse sciatic nerves than in wt (since we are dealing with CMT1 and not CMT2 defects)?
As for PMP22 presence in CA3 hippocampal neuron nuclei, how would that interfere with the cell cycle? (besides, the 3-star statistical difference between wt and TrJ CA3 neurons is hard to see in Fig.6).
While some therapeutical approaches for PMP22-based CMTs are being developed, do the authors think that a -potential- implication of the protein in cell cycle regulation could be interesting and if yes, how?
Specific points:
The ms will greatly profit from careful re-reading (if possible by a native speaker; examples in the Abstract: PMP22 mutations were are pathognomonic; ...the same point mutations of as CMT1E patients...). Many small typing and other errors throughout (CTM instead of CMT, ...).
- For all figures with statistics plots, the number of evaluated items should be mentioned in the legend. We know that 10+10 mice were used, but for measures of nuclear volumes etc. the number of nuclei analyzed per condition is important, especially since S.E.M. bars often seem quite high. In M&M section, the sentence "each experimental procedure was carried out in triplicate" is not clear to me. Did the authors find significant differences from one animal to another? "± 200 nuclei per phenotype were included in each analysis", is that ~20 nuclei per animal? An overview image of teased sciatic nerves from wt and mutated animals would have been welcome. Finally, personally I would have appreciated one "summary sentence" for each figure telling what message we should get.
- "DAPI-dark" is easily misleading, as it denotes actually the strongly fluorescent zones appearing thus "light" (bright) under the microscope (=condensed HC)?
- In Figs. 1 and 3-5, judging from the scale bars, all fibers would be 20-30µm in diameter, and the SC nuclei 60µm long/25µm large? It's possible, but seems rather big to me.
- Figure 1 legend: B-C should read B-D. The labeling of the y-axis is hardly readable even at 300% magnification, and there are several errors (ciatic, volumen).
- Line 467 mentions a Figure 1E that doesn't exist.
- Figure 2: it is not clear to me how exactly DAPI and H3K4m23 overall intensities were measured and compared, since TrJ nuclei are smaller than wt; how was this taken into account? (even if TrJ nuclei are smaller, they should contain the same amount of DNA, maybe transcriptionally less active/more condensed). Were pixel (voxel) intensities summed up over the whole nuclear volume (z-stack), then divided by the volume? This is also not very clear for me from the M&M section. Furthermore, were the wt and TrJ stainings done together, at best on the same slide? (immunostaining total intensities may come out differently from one experiment to another, especially when one wants to compare two or three antigens for their relative amounts).
- Figure 3: so PMP22 signal is more associated with peripheral regions where there is less chromatin; what do the authors make of this observation?
- Figure 4: legend: (Entire-C) should read TC (total-chromatin) as in the figure and in the text (where Entire-C is later also found). Figure 4 meaning is not clear to me; it seems logical that chromatin is a part of total chromatin, but fraction M2 (part of total chromatin in euchromatin) ?? Also, Figure 4C is strange for the M1&M2 values where there is no bars. Finally, how is it possible that EC colocalization with TC is stronger in wt than in TrJ nuclei (lines 550-52)? All EC should be part of TC, no? Is this a threshold problem?
- Figure 5: From the chosen example images it is hard to see that PMP22 colocalizes to 80% with EC, but almost not with HC, particularly for the wt nucleus (maybe because the analysis threshold was set differently for DAPI-dark and EC zones, or because we don't see the z-stack?). Maybe an additional double-labeled image may have been interesting (green&red=yellow). And again, as PMP22 signal is found mostly in chromatin-poor regions (Fig. 3), how does this go together?
- Figure 6: the fibers in CA3 (particularly wt) are quite strongly stained with PMP22 antibody, more clearly than the nuclei?

Author Response
Thank you for your careful review. We appreciate the extensive comments, some of which would also merit direct exchange of views, and approaches. Below we transcribe your questions and/or suggestions followed by our answers:
Reviewer: Overall, the reviewer sees like an "imbalance" between reported facts and length of the paper. Much of the ms seems more of a review, than a research article (almost 3 print pages of Introduction, and a large part of the Discussion section dealing with findings not directly related to the present work, but rather general knowledge about PMP22).
Answer: We considered your appreciation and shortened the text in all sections of the new manuscript version, trying to avoid general knowledge in the Discussion section.
Reviewer: The title: "The PMP22 myelin protein is expressed in Schwann cell nuclei": the nuclear presence of PMP22 in SC is not really a new fact, as the authors themselves already described it for ex. in their publication Kun et al. 2012 (book chapter; see e.g. Fig.4 in that work), DOI: 10.5772/35306. More on this later.
Answer: Our first findings of the nuclear presence of PMP22 were descriptive (Kun et al., 2012)[51], indicating the presence of both the transcript and the protein, more noticeable in the SC nuclei of TrJ than in those of Wt. In that work, we did not perform any quantification that would allow us to evaluate the levels of PMP22 both in TrJ and in Wt. However, due to the frequency with which this event occurred, it seemed pertinent to communicate the finding. The work of Rosso et al. (2012)[58], also showed the presence of nuclear PMP22 but focused on quantifying perinuclear aggregates of PMP22. However, those approaches did not include a comparative quantification of the relationship between PMP22 and nuclear subcompartments with different transcriptional levels or nucleoskeleton structures. Furthermore, in the new version of the manuscript, we also present data on Lamina B1, not published in the literature to date. It is clear that the results of this study come from the deepening of the first data published in 2012. To the best of our knowledge, the results presented in the current manuscript are original and, for these reasons, valuable.
Reviewer: The rationale for the research is contained in Introduction, lines 118-140. In my eyes, the argumentation falls a bit short; thus, current research does not consider that demyelination in CMT1E is due only to misfolding of mutated PMP22 and toxic aggregation with normal PMP22 affecting membrane transport and degradation. For example, already a slight stoichiometric imbalance between the different myelin components (PMP22 interacts with P0) is fatal for myelin formation and maintenance, among still other causes implying PMP22 protein.
Answer: We agree with you. There was only a partial mechanism involved. Stoichiometric imbalance and another putative not known mechanism would be implicated. Therefore, we decided not to mention the different pathophysiological mechanisms in the Introduction section of the new manuscript version.
Reviewer: The authors then argue that, to explain CMT disease onset, a potential role of PMP22 in cell cycle control has so far not been taken into account "as adjunctive cause", although the gene had originally been discovered as gas-3 (growth arrest) gene in NIH3T3 fibroblasts. This argument is however questionable since in fact, a role for PMP22 in cell survival and/or proliferation was suggested very early, some of the corresponding literature is even cited in the ms (ex., refs. [5], [78]). In 2005, Giambonini-Brugnoli et al. (not cited) write: "Our findings [...] provide the first in vivo evidence for a direct role of PMP22 in the regulation of cell proliferation." And later: "Increased expression of genes involved in cell cycle regulation and DNA replication is characteristic and specific for the early stage in Pmp22 mice, supporting a primary function of PMP22 in the regulation of Schwann cell proliferation. In the Tr mutant, a distinguishing feature is the high expression of stress response genes..." (Neurobiology of Disease 18:656-68). In human SC, PMP22 was reported to function as growth arrest gene acting on multiple signaling pathways and transcription factors like Knox20, EGR2, Sox10 (see for ex. ref. [19]: Li et al. 2013, Mol Neurobiol 47(2):673-98; a very detailed review on all aspects of PMP22 and CMT). Is there a clear difference between the proposal of these former studies, and that of the present manuscript?
Answer: In the previous manuscript, we presented a hypothesis based on our current data, our preliminary results, and findings from the scientific literature, which we describe in one of our answers below. We believe that we failed to adequately explain the premises that led to the formulation of our assumption. We also made it in the light of prior knowledge. We regret not having detected the work of Giambonini-Brugnoli et al. (2005) when reviewing the literature, for which we appreciate the information and included this reference in the new version.
Reviewer: Concerning the content of the ms, while the immunocytochemical part and in particular, the colocalization studies are well conducted (with quite sophisticated image and statistical analyses making up for a big part of the ms), the reviewer regrets that the theory advanced by the authors is entirely based on "simple" fluorescence staining of SC nuclei with a single antibody. This falls even behind the above-cited work of 2012 where IHS was also employed to detect pmp22 RNA in the nucleus. By the way it would have been interesting to discuss the difference between the results reported in 2012 and in the present ms: in the 2012 paper PMP22 protein had been detected only in TrJ, not in wt nuclei (different antibodies, different antigen retrieval protocols, different microscopy, artifact?).
Answer: Our preliminary findings of the nuclear presence of PMP22 were only descriptive, indicating that both transcripts and protein accumulation were higher in SCs of TrJ than in Wt. At that time, no attempt was made to specifically analyze nuclear expression between Wt and TrJ. It was for this reason that we carried out this work. We think that the difference in the results could be related to the magnification with which the images were taken (20x and 40x) and the lower expression of PMP22 in the Wt SC nuclei than in TrJ. In the present work, we analyze the nuclei with the highest magnification (63x) for better visualization and quantification.
Reviewer: When the authors claim "mutated PMP22 was more colocalized with heterochromatin than normal PMP22" (Abstract line 30, and elsewhere): It is unlikely that their antibody staining approach can distinguish between wt and mutated PMP22 protein. Moreover, since the TrJ mouse is heterozygous for the mutation, can we only be sure that the nuclear expression concerns indeed the mutated protein? It would have been interesting to know whether the higher presence of PMP22 in TrJ nuclei was essentially due to mutated protein, or wt, or maybe even a complex/aggregate of both? Is the mutated gene generally (not only in the nucleus) upregulated in the cell to make up for lack of normal PMP22 protein (which may have as corollary a higher transport into the nucleus)? Nuclei are rather easy to purify by centrifugation, so even Western blotting or other analysis methods could be employed.
Answer: We have used two different anti-PMP22 antibodies, not observing differences between the immunostaining of both antibodies:
(1) ab61220 from Abcam, a rabbit polyclonal antibody, which recognizes amino acids 111-160, including an extensive protein domain containing the terminal carboxyl region, and
(2) sc65739 from Santa Cruz, a mouse monoclonal antibody that recognizes amino acids 121 to 123 of the second extracellular domain of the protein.
Neither of these two antibodies recognizes the first transmembrane domain where the TrJ mutation is located (position 16). Therefore, the signals obtained recognized the presence of PMP22 without differentiating whether or not the protein carried the mutation. Thus, we agree with you and have corrected this semantic error, which changed the meaning of the sentence, as you rightly point out. However, it is important to clarify the rationale for our approach when using formic acid unmasking. To the best of our knowledge, formic acid unmasking, a procedure widely used to characterize the intracellular components of β-amyloid in the diagnosis and study of Alzheimer's disease, assumes that exposure of the main aggregated components of the cell is obtained ( Kitamoto et al., 1987, Wisniewski et al, 1989; Kun et al., 2018). In our case, we have optimized the procedure for peripheral nerve fibers and mouse brain (Damián et al., 2021) and we assumed that the previous treatment with formic acid mainly eliminated the non-aggregated content, within which a high percentage of PMP22 (among other proteins) is found. Therefore, using the unmasking process, the aggregated component mainly containing the mutated protein was mainly identified.
- Kitamoto T, Ogomori K, Tateishi J, Prusiner SB (1987). Formic acid pretreatment enhances immunostaining of cerebral and systemic amyloids. Lab Invest 57(2): 230-236.
- Wisniewski HM, Wen GY, Kim KS (1989). Comparison of staining methods on the detection of neuritic plaques. Acta Neuropathol 78(1):22-7.
- Kun A, Gonzáles-Camacho F, Hernández S, Moreno-García A, Calero O, Calero M (2018). Characterization of amyloid-β plaques and autofluorescent lipofuscin aggregates in Alzheimer's disease brain: A confocal microscopy approach. In: Chapter 3: Amyloid Methods and Protocols (3th Ed, Sigurdsson EM, Calero M, Gasset M Eds.) Methods Mol Biol 1779, pp. 497-412.
Reviewer: Thus, in the end, the scientific facts on which the authors have based their theory (cell cycle regulation by PMP22, defective if mutated) appear a bit weak, based on rather indirect evidence and statistical correlation/coincidence. The authors themselves seem not sure how to interpret their data, resp. how to put them into the context of their theory; at the end of the "Conclusion" section we read: "The prevalent PMP22 location at peripheral nuclear regions of both genotypes could indicate its access and level of diffusion into the nucleus or a possible association of PMP22 with nuclear lamina or with active or inactive chromatin associated with the nuclear lamina". This seems a bit vague and it is not self-evident that these observations are indicative of a cell cycle regulation role for PMP22.
Answer: What the Reviewer mentions also does not represent for us evidence of the role of cell cycle deregulation by PMP22 mutation.
Our thinking was based on our current data, our preliminary results, and reported findings in the scientific literature.
We summarized our findings below:
- The location of PMP22 into SC nuclei of Wt and TrJ sciatic nerves fiber
- The greater amount of PMP22 expressed in CS nuclei of TrJ than Wt
- The colocalization of PMP22 with transcriptional competent EC in both Wt and TrJ genotypes, but with more colocalization in the normal Wt
- The lesser colocalization of PMP22 with de silent HC than the EC in both genotypes, however with slightly more PMP22/HC colocalization in the mutated TrJ
- The higher number of CS nuclei observed in sciatic nerves of TrJ regarding Wt
These results integrate with others reported in the literature to shed light on the complex functions of PMP22. In this sense, we would like to highlight the following contributions:
- The higher amount of PMP22 transcripts in SC nuclei and cytoplasm of TrJ regarding Wt, reported previously by us [51,58]
- The previously proposed role for pmp22 as a growth arrest-specific (gas3) which is up-regulated by serum starvation (G0 cell cycle) [5]
- The negative gene modulation of SC cell proliferation by PMP22 [43]
- The requiring SC differentiation to achieve the myelination [40-42]
- Our preliminary results obtained in three TrJ and three Wt mice that showed greater incorporation of BrdU in TrJ CS nuclei than in Wt (data not shown in the present manuscript).
- The altered myelination of TrJ and CTM1E disease [31,32].
The presence of intranuclear PMP22 and its association with the transcriptionally competent chromatin context, mainly in the normal genotype, suggests the nuclear incorporation of PMP22 to fulfill a conceivable nuclear function. The role previously proposed for pmp22 as a specific growth arrest gene and as a negative modulator gene for cell proliferation suggest that PMP22 could contribute to maintaining SC in G0, inhibiting its proliferation to allow SC differentiation, which is necessary to achieve the myelination process. This function of PMP22 could be partially affected in TrJ SC nuclei. We speculate that TrJ SC nuclei carrying mutated pmp22 would fail to maintain the differentiated state, and continue to proliferate. This undifferentiated SC state would affect nerve myelination. The SC would attempt to compensate for impaired PMP22 function by transcribing, synthesizing, and incorporating more PMP22 into the SC nuclei of TrJ than Wt. New approaches will be needed to unravel the functions of this protein. Our findings open up a field of future research, which we are beginning to delve into.
Reviewer: At the same time, quite many PMP22-defective mouse models are available today. The reviewer thinks that already a similar antibody staining of those mice could probably contribute to verify the authors' theory: what happens to nuclear staining when PMP22 is under- or over-expressed to different degrees; and how is this correlated to SC cell cycle?
Answer: We think that what is raised by the Reviewer is interesting and can constitute a valuable future approach.
Reviewer: While it is in line with the authors' theory that the number of SC nuclei is higher in TrJ mice, is there an explanation why there seem to be also more fibers in TrJ mouse sciatic nerves than in wt (since we are dealing with CMT1 and not CMT2 defects)?
Answer: We have not performed a comparative quantification of the total number of fibers in intact sciatic nerves of Wt with respect to TrJ. Although we believe that this would be an interesting study to elucidate the neuronal, sensory, and motor, involvement in peripheral neuropathy modeled in TrJ. Explaining this fact implies the study of neurogenesis in the peripheral nervous system in normal and pathological conditions, but this is not currently within our research horizon. It is important to clarify that the data on the number of sciatic nerve fibers were obtained by analyzing images of TrJ fibers in focal planes covering areas equivalent to those analyzed in Wt. All images were acquired at the same magnification and with the fibers completely covering the focal plane. Regarding the number of nuclei in each genotype, they have been expressed as the number of nuclei per fiber, to make it independent of the number of fibers.
Reviewer: As for PMP22 presence in CA3 hippocampal neuron nuclei, how would that interfere with the cell cycle? (besides, the 3-star statistical difference between wt and TrJ CA3 neurons is hard to see in Fig.6).
Answer: There is evidence that PMP22 mRNA and protein are expressed in other CNS regions different than CA3 (Spreyer et al., 1991; Welcher et al., 1991; Bosse et al. 1994; Parmantier et al., 1995; 1997) showing lower expression than in PNS. We recently found that both mRNA and protein were expressed in hippocampal neurons (Damian et al., 2021). Results presented in Figure 6 demonstrating a higher level of PMP22 in SC than in CA3 nuclei could depend on the lower global expression of PMP22 in CNS than in PNS. We speculate that at the central level, PMP22 could exert at least the same nuclear function that it fulfills at the peripheral level associated with cell cycle arrest (Manfioletti et al., 1990) [5].
However, we have decided to withdraw the CNS results to present them together with other analyzes in a future manuscript.
- Spreyer P, Kuhn G, Hanemann CO, Gillen C, Schaal H, Kuhn R, Lemke G, Müller HW (1991). Axon-regulated expression of a Schwann cell transcript that is homologous to a ‘growth arrest-specific gene. EMBO J 10:3661–3668.
- Welcher AA, Suter U, De Leon M, Snipes GJ, Shooter EM (1991). A myelin protein is encoded by the homolog of a growth arrest-specific gene. Proc Natl Acad Sci USA 88:7195–7199.
- Bosse F, Zoidl G, Wilms S, Gillen CP, Kuhn HG, Müller HW (1994). Differential expression of two mRNA species indicates a dual function of peripheral myelin protein PMP22 in cell growth and myelination. J Neurosci Res 37:529–537.
- Parmantier E, Cabon F, Braun C, D'Urso D, Muller HW, Zalc B (1995). Peripheral myelin protein-22 is expressed in rat and mouse brain and spinal cord motoneurons. Eur J Neurosci.7:1080–1088.
- Parmantier, E, Braun C, Thomas JL, Peyron F, Martinez S, Zalc B (1997). PMP-22 expression in the central nervous system of the embryonic mouse defines potential transverse segments and longitudinal columns. J Comp Neurol 378: 159–172.
- Damián JP, Vázquez Alberdi L, Canclini L, Rosso G, Bravo S O, Martínez M, Uriarte N, Ruiz P, Calero M, Di Tomaso MV, Kun A (2021). Central alteration in peripheral neuropathy of Trembler-J mice: Hippocampal pmp22 expression and behavioral profile in anxiety tests. Biomolecules 11: 601-617.
Reviewer: While some therapeutical approaches for PMP22-based CMTs are being developed, do the authors think that a –potential implication of the protein in cell cycle regulation could be interesting and if yes, how?
Answer: Dysregulation of the cell cycle with the persistence of non-oncogenic cell proliferation is an aspect that deserves an in-depth study since it would be a process of dysregulation with cell proliferation different from the one that occurs during tumor development. In turn, their knowledge could allow different therapeutic approaches to peripheral neuropathies and even neoplastic diseases in the future.
We would like to point out some considerations that allowed us to contextualize our approach. PMP22 is a small, highly hydrophobic protein that, under normal conditions, has chaperone-assisted folding, with a high propensity to aggregate forming perinuclear aggresomes. More than 80% of newly synthesized PMP22 has difficulty in inserting into the membrane and is rapidly degraded by the proteasome (Notterpek et al., 1999; Parek et al, 1997). Although PMP22 is a ubiquitous protein, it is in SC where it has the highest expression levels (Li et al, 2013). PMP22 profoundly affects SCs, especially due to problems related to folding, insertion, degradation, and the formation of metabolically active aggregates (Rangaraju et al, 2009). The functional cost of myelin homeostasis with correct integration of PMP22 is very high, given that only 20% of the protein synthesized under normal conditions is inserted into the membrane. This implies a high energy expenditure not only in the synthesis of the protein that will not finally be integrated into the membrane but also in the maintenance of the proteasomal machinery responsible for the clearance of the excess protein not integrated into the membrane. Why is this "expensive" protein still being synthesized? What is the “benefit” offered by their roles? The nuclear presence of PMP22 and its early characterization as GAS3 add complexity to the understanding of its possible functions, which also occurs with other claudin family proteins, such as PMP22, catenins. However, although its expression is higher in G0, the pmp22 gene is expressed in all stages of the cell cycle, indicating other possible roles, with an influence on cell homeostasis. For example, some authors suggest that PMP22 and PERP (another protein of the PMP22/GAS3 family) participate in a balance that involves regulation of the cell cycle by PMP22 and increases apoptosis by PERP, via p53 (Attardi et al., 2000; Hou et al., 2021). Cell cycle control is integrated into myelinating CS with other transcendental events such as myelination itself and inhibition of axonal sprouting and both are supported by the transcellular dialogue between glia and neurons, for the maintenance of homeostasis of the peripheral nerve fiber. PMP22 could be playing a crucial role (and in this sense, structurally and functionally "transversal") in the synchronization (alone or as a component of supramolecular, nuclear, or cellular complexes) of at least two of these important events in CS, cell arrest and myelination, to which the balance between maintenance in G0/apoptosis or another unknown function could eventually be incorporated.
In this sense, elucidating the mechanism of entry of PMP22 into the nucleus could probably lead to revealing an eventual association of PMP22 with other proteins that negatively could modulate the cell cycle.
- Notterpek L, Ryan MC, Tobler AR, Shooter EM (1999). PMP22 accumulation in aggresomes: Implications for CMT1A pathology. Neurobiol Dis 6: 450–460
- Pareek S, Notterpek L, Snipes GJ, Naef R, Sossin W, Laliberté J, Iacampo S, Suter U, Shooter EM, Murphy RA (1997) Neurons Promote the Translocation of Peripheral Myelin Protein 22 into Myelin. Neurosci 17(20):7754-7762.
- Li J, Parker B, Martyn C, Natarajan C, Guo, J (2013). The PMP22 gene and its related diseases. Mol Neurobiol 47: 673–698
- Rangaraju S, Hankins D, Madorsky I, Madorsky E, Lee W-H, Carter CS, Leeuwenburgh C, Notterpek L (2009). Molecular architecture of myelinated peripheral nerves is supported by calorie restriction with aging. Aging Cell 8(2):178-191.
- Attardi LD, Reczek EE, Cosmas C, Demicco EG, McCurrach ME, Lowe SW, Jacks T (2000). PERP, an apoptosis-associated target of p53, is a novel member of the PMP-22/gas3 family. Genes Dev 14:704–718.
- Hou J, Wang L, Zhao J, Zhuo H, Cheng J, Chen X, Zheng W, Hong Z, Cai J (2021). Inhibition of protein PMP22 enhances etoposide-induced cell apoptosis by p53 signaling pathway in Gastric Cancer. Int J Biol Sci 17(12): 3145–3157.
Specific points:
Reviewer: 1-The ms will greatly profit from careful re-reading (if possible by a native speaker; examples in the Abstract: PMP22 mutations were are pathognomonic; ...the same point mutations of as CMT1E patients...). Many small typing and other errors throughout (CTM instead of CMT, ...). –
Answer: The English writing of the new manuscript was revised.
Reviewer: 2- For all figures with statistics plots, the number of evaluated items should be mentioned in the legend. We know that 10+10 mice were used, but for measures of nuclear volumes, etc. the number of nuclei analyzed per condition is important, especially since S.E.M. bars often seem quite high. In the M&M section, the sentence "each experimental procedure was carried out in triplicate" is not clear to me. Did the authors find significant differences from one animal to another? "± 200 nuclei per phenotype were included in each analysis", is that ~20 nuclei per animal? An overview image of teased sciatic nerves from wt and mutated animals would have been welcome. Finally, personally I would have appreciated one "summary sentence" for each figure telling what message we should get. –
Answer: 2- Each experimental procedure was performed jointly in Wt and TrJ mice and was also implemented in triplicate. Wt and TrJ sciatic nerves were processed, immunostained, and observed with confocal microscopy in parallels. Medians (each corresponding to one outcome) were compared with each other and data were pooled when p-values were ≥ 0.05. All slices from the z-stack images of SC nuclei were included to measure the mean intensity of the different signals, and the 10 central slices of each nucleus were used to accomplish the colocalization analyses. The numbers of Wt and TrJ SC nuclei included in each analysis are presented in the corresponding Figure capture of the new manuscript version. This information was introduced in the Material and Methods section of the manuscript.
A summary sentence about the results was presented in each Figure capture.
As was requested, the attached Figure 1 shows an overview image of teased sciatic nerves from Wt and TrJ is presented. The following image was obtained during the optimization procedures of the BrdU incorporation and immunodetection technique (fibers marked in red). Nuclei were identified by DAPI (green). BrdU positive nuclei appear in yellow (white asterisk). Green bar: 10 μm
Reviewer: 3- "DAPI-dark" is easily misleading, as it denotes actually the strongly fluorescent zones appearing thus "light" (bright) under the microscope (=condensed HC)?
Answer: 3- Yes, this is true. To analyze differences between DAPI-light and DAPI-dark regions, we define a signal intensity threshold in DAPI channel, by visual inspection in each nucleus image (resolution: 1024 x 1024 pixels, pixel size 0.1 x 0.1microns, 63x lens, 2x zoom), using FIJI software Thus, we established the pixel intensity value threshold to categorize as DAPI-light the EC and as a DAPI-dark the HC. The threshold was set at < 150 intensity arbitrary units for assigning DAPI-light euchromatic pixels and ≥ 150 intensity arbitrary units for DAPI-dark heterochromatic pixels. The intensity fluorescence range belonging to DAPI-light was comprised between ≥ 50 to <150, and to DAPI-dark, between ≥150 to 255. The above description is now included in the Materials and Methods section of the new manuscript.
Reviewer: 4- In Figs. 1 and 3-5, judging from the scale bars, all fibers would be 20-30µm in diameter, and the SC nuclei 60µm long/25µm large? It's possible, but seems rather big to me.
We especially thank the Reviewer for this question because, taking into account your comment, we have carefully checked the relationship between magnification and bars length of all the Figures. Effectively, we have found a systematic error in the transference Fiji/Photoshop programs, which we have now corrected. Considering your question, we mentioned that the SC nuclei of the Wt genotype are long (6-8 µm approx.) and well packed, appearing parallel to the major axis of the teased sciatic nerve fiber whole mount. Under the same condition, TrJ SC nuclei have a slightly more spherical appearance and lesser longer (4- 6 µm approx.). The difference in the nerve fiber spatial arrangements of SC nuclei between TrJ and Wt could be the consequence of a deficiency in the nerve fibers packing due to the absence of collagen IV in the basal lamina of the TrJ fibers (Rosso et al., 2012) [59].
Reviewer: 5- Figure 1 legend: B-C should read B-D. The labeling of the y axis is hardly readable even at 300% magnification, and there are several errors (ciatic, volumen). Line 467 mentions a Figure 1E that doesn't exist.
Answer: 5- “B-C” of Figure 1 was substituted by “B-D”. We wrote the content of the y-axes with a larger font and corrected all pointed errors.
At 467 line, “Figure 1 E” was removed and substituted by “Figure 1 D”, however, the corresponding line number is different in the new manuscript version
Reviewer: 6- Figure 2: it is not clear to me how exactly DAPI and H3K4m23 overall intensities were measured and compared, since TrJ nuclei are smaller than wt; how was this taken into account? (even if TrJ nuclei are smaller, they should contain the same amount of DNA, maybe transcriptionally less active/more condensed). Were pixel (voxel) intensities summed up over the whole nuclear volume (z-stack), then divided by the volume? This is also not very clear for me from the M&M section. Furthermore, were the wt and TrJ stainings done together, at best on the same slide? (immunostaining total intensities may come out differently from one experiment to another, especially when one wants to compare two or three antigens for their relative amounts).
Answer: 7- All FIJI plugins that measure signal intensities provided in arbitrary units the mean intensity for each measurement that means: fluorescence intensity/nuclear area. Using FIJI, we obtained the mean intensity values in all z-stack of each SC nuclear image. Therefore, since we perform intensity/area measurements in each image plane, we cover, plane by plane, the entire nuclear volume. In this way, the differences in nuclear size between TrJ and Wt were taken into account in each performed measurement. We explained this issue in the corrected manuscript.
Each experimental procedure was performed jointly in Wt and TrJ mice and implemented in triplicate. The sciatic nerves of Wt and TrJ were in different slices, but were processed, immunostained, and analyzed with the confocal microscopy in parallel.
The medians of each of the variables of each experiment were calculated and compared with each other. Thus, data were pooled when comparison p-values were ≥ 0.05, indicating no differences between results from triplicates.
Reviewer: 7- - Figure 3: so PMP22 signal is more associated with peripheral regions where there is less chromatin; what do the authors make of this observation?
In most the eukaryotic nuclei, the condensed heterochromatin and the repressive epigenetic histone modifications (including H3K9me2,3 and H3K27me2,3) are enriched at the nuclear periphery (Wu et all., 2005; Reik et al., 2007; Eberhart et al., 2013; Lafon-Hughes et al., 2013). We observed that the SC nuclei differentiated to the canonical nuclear organization since its nuclear periphery is not enriched in DAPI-dark signals. Furthermore, both heterochromatic and euchromatic signals have lesser intensity in the periphery (Figure 3). Therefore, we think that it could be a region with lesser chromatin adjacent to the nuclear lamina. It is in this region where PMP22 and Lamin B1 signals predominate and colocalize (Figure 7).
- Wu R, Terry AV, Singh PB, Gilbert DM (2005). Differential subnuclear localization and replication timing of Histone H3 lysine 9 methylation states. Mol Biol Cell 16: 2872–2881.
- Reik W (2007). Stability and flexibility of epigenetic gene regulation in mammalian development. Nature, 447, 425–432.
- Eberhart A, Feodorova Y, Song C, Wanner G, Kiseleva E, Furukawa T, Kimura H, Schotta G, Leonhardt H, Joffe B, Solovei I (2013). Epigenetics of eu- and heterochromatin in inverted and conventional nuclei from mouse retina. Chromosome Res 21: 535–554.
- Lafon-Hughes L, Di Tomaso MV, Liddle P, Reyes-Ábalos AL,Folle GA (2013). Preferential localization of gammaH2AX foci in euchromatin of retina rod cells after DNA damage induction. Chromosome Res 21: 789 - 803.
Reviewer: 8- - Figure 4: legend: (Entire-C) should read TC (total-chromatin) as in the figure and in the text (where Entire-C is later also found). Figure 4 meaning is not clear to me; it seems logical that chromatin is a part of total chromatin, but fraction M2 (part of total chromatin in euchromatin) ?? Also, Figure 4C is strange for the M1&M2 values where there is no bars. Finally, how is it possible that EC colocalization with TC is stronger in Wt than in TrJ nuclei (lines 550-52)? All EC should be part of TC, no? Is this a threshold problem?
Answer: 8- “Entire-chromatin” was substituted by “total-chromatin” (TC) in all the text.
With respect to Figure 4 (Figure 5 in the new manuscript version) there are no bars because the median of M1 and M2 coefficients are equal to 0 (M1 = 0 and M2 = 0) indicating that there is no colocalization between EC and heterochromatin (HC) in both genotypes.
With respect to your question, we reason that if euchromatin (EC) is a part of TC (M1), it is logical that a part of the TC corresponds to EC (complementary M2). It can be illustrated in the attached Figure 2.
We believe that the result related to a stronger colocalization of EC with TC in SC nuclei of Wt than TrJ is not a threshold issue. In both cases, we applied the same threshold to differentiate DAPI-dark HC from DAPI-light EC, and additionally we identified EC using the anti-H3K4m3 antibody. Considering the lower global intensity of the DAPI signal and the higher intensity of the H3K4m3 mark in SC nuclei of Wt relative to TrJ, we think that SC Wt nuclei harbor more EC than SC nuclei of TrJ. This could be the reason why a greater colocalization was obtained between EC and CT.
Reviewer: 9- Figure 5: From the chosen example images it is hard to see that PMP22 colocalizes to 80% with EC, but almost not with HC, particularly for the wt nucleus (maybe because the analysis threshold was set differently for DAPI-dark and EC zones, or because we don't see the z-stack?). Maybe an additional double-labeled image may have been interesting (green&red=yellow). And again, as PMP22 signal is found mostly in chromatin-poor regions (Fig. 3), how does this go together?
Answer: 9- The colocalization results were obtained over 100 nucleus images, analyzing 10 central slices in each. Therefore, the obtained results are robust. If we consider the way colocalization is calculated, we understand that a value of 0.8 for M1 or M2 does not mean 80% colocalization between signals. The coefficient values range between 0 (no colocalization) and 1 (colocalización). Therefore, 0.8 represents a high value of the coefficient but does not mean that 80% of the PMP22 area is collocated with EC or vice versa. Furthermore, not all nuclei have exactly the same colocalization coefficient values. The graph in Figure 5 (Figure 6 in the new version of the manuscript) illustrates the median of the colocalization coefficients of Wt and TrJ SC nuclei.
As requested, we have incorporated double-labeled images in Figures 5, 6, and 7 that illustrate colocalization.
Reviewer: 10- Does colocalization mean interaction?
In many research articles, fluorescence colocalization is used to state that two molecules of interest interact with each other. While a functional connection might be plausible in case of a clear colocalization, a real physical interaction can certainly not be claimed based on a mere colocalization.
Answer: 10- We also consider that while a functional connection might be plausible a real physical interaction cannot be claimed based on a mere colocalization. It needs further approaches, such as Förster resonance energy transfer (FRET) a tool used for analyzing spatiotemporal localization of molecular interactions. FRET consists of the analysis of the non-radiative energy transfer that occurs between two fluorophores (donor and acceptor) in which the emission spectrum of the donor and the excitation spectrum of the acceptor overlap sufficiently, so that the donor, once excited, transfers energy to the acceptor which subsequently fluoresces. Since the efficiency of FRET decays with the sixth potency of the distance between fluorophores (Forster radius) the physical interaction between two molecules could be estimated. The Förster radius is referred to as the distance at which the efficiency of energy transfer is 50%. The distance over which the energy can be transferred is dependent on the spectral characteristics of the fluorophores but is generally in the range of 10–100 Å.
- Sanders JC, Holmstrom ED (2021). Integrating single-molecule FRET and biomolecular simulations to study diverse interactions between nucleic acids and proteins Joshua C. Essays Biochem 65: 37–49.
